

# MICS-Asia III: Multi-model comparison and evaluation of aerosol over East Asia

Lei Chen[1,2], Yi Gao[1], Meigen Zhang[1,3,4], Joshua S. Fu[5], Jia Zhu[6], Hong Liao[2,7], Jialin Li[1], Kan Huang[5], Baozhu Ge[1], Xuemei Wang[8], Yun Fat LAM[9], Chuan Yao Lin[10], Syuichi Itahashi[11,12], Tatsuya Nagashima[13], Mizuo Kajino[14,15], Kazuyo Yamaji[16], Zifa Wang[1,3], Jun-ichi Kurokawa[17]

[1]State Key Laboratory of Atmospheric Boundary Layer Physics and Atmospheric Chemistry, Institute of Atmospheric Physics, Chinese Academy of Sciences, Beijing, China
[2]School of Environmental Science and Engineering, Nanjing University of Information Science & Technology, Nanjing 210044, China
[3]University of Chinese Academy of Sciences, Beijing, China
[4]Center for Excellence in Regional Atmospheric Environment, Institute of Urban Environment, Chinese Academy of Sciences, Xiamen, China
[5]Department of Civil and Environmental Engineering, University of Tennessee, Knoxville, TN 37996, USA
[6]Research Institute of Climatic and Environmental Governance, Nanjing University of Information Science & Technology, Nanjing 210044, China
[7]International Joint Laboratory on Climate and Environmental Change, Nanjing University of Information Science & Technology, Nanjing 210044, China
[8]Institute for Environment and Climate Research, Jinan University, Guangzhou, China
[9]School of Energy and Environment, City University of Hong Kong, Hong Kong
[10]Research Center for Environmental Changes, Academia Sinica, Taiwan
[11]Central Research Institute of Electric Power Industry, Abiko, Chiba 270-1194, Japan;
[12]Department of Marine, Earth, and Atmospheric Sciences, North Carolina State University, Raleigh, NC 27607, USA
[13]National Institute for Environmental Studies, Tsukuba, Japan
[14]Meteorological Research Institute, Japan Meteorological Agency, Tsukuba, 305-0052, Japan
[15]Faculty of Life and Environmental Sciences, University of Tsukuba, Tsukuba, 305-8577, Japan
[16]School of Science and Engineering, Meisei University, Hino, Tokyo 191-8506, Japan
[17]Asia Center for Air Pollution Research, 1182 Sowa, Nishi-ku, Niigata, Niigata, 950-2144, Japan

*Correspondence to*: M.G. Zhang (mgzhang@mail.iap.ac.cn)

**Abstract.** Fourteen chemical transport models (CTMs) participate in the MICS−Asia Phase III Topic 1. Their simulation results are compared with each other and with an extensive set of measurements, aiming to evaluate the current multi−scale air quality models' ability in simulating aerosol species and to document similarities and differences among model performances, also to reveal the characteristics of aerosol chemical components over big cities in East Asia. In general, all participant models can reproduce the spatial distribution and seasonal variability of aerosol concentrations in the year 2010, and multi−model ensemble mean (EM) shows better performance than most individual models, with Rs ranging from 0.65 ($NO_3^-$) to 0.83 ($PM_{2.5}$). Underestimations of BC (NMB=−17.0%), $SO_4^{2-}$ (NMB=−19.1%) and $PM_{10}$ (NMB=−32.6%) are simulated by EM, but positive biases are shown in $NO_3^-$ (NMB=4.9%), $NH_4^+$ (NMB=14.0%) and $PM_{2.5}$ (NMB=4.4%). Simulation results of BC, OC, $SO_4^{2-}$, $NO_3^-$ and $NH_4^+$ among CTMs are in good agreements, especially over polluted areas,



such as the eastern China and the northern part of India. But large coefficients of variations (CV > 1.5) are also calculated over arid and semi–arid regions. This poor consistency among CTMs may attribute to their different processing capacities for dust aerosols. According to the simulation results in the six Asian cities from EM, different air–pollution control plans should be made due to their different major air pollutants in different seasons. Although a more considerable capacity for

5    reproducing the concentrations of aerosol chemical compositions and their variation tendencies is shown in current CTMs by comparing statistics (e.g. RMSE and R) between MICS–Asia Phase II and Phase III, detailed process analysis and a fully understanding of the source–receptor relationship in each process may be helpful to explain and to reduce large diversities of simulated aerosol concentrations among CTMs, and these may be the potential development directions for future modeling studies in East Asia.





## 1 Introduction

Rapid urbanization and industrialization have stimulated economic growth and population expansion during the last several decades in East Asia (Spence et al., 2008; Yan et al., 2016; Chen et al., 2016), but also brought about the noticeable degradation of ecological environment at the same time (Hall 2002; Han et al., 2014; Yue et al., 2017). Significant increase in atmospheric aerosol loading, especially from anthropogenic emissions, can exert much influence on weather (Cowan et al., 2013; Gao et al., 2015a), climate (Wang et al., 2016a), air quality (Gao et al., 2016a), and even human health (Carmichael et al., 2009). For example, aerosols can enhance the absorption and scattering of solar radiation to modify the thermodynamic structure of the atmospheric boundary layer (Ding et al., 2016; Petaja et al., 2016), can act as cloud condensation nuclei and ice nuclei to alter cloud properties and precipitation (Lohmann and Diehl, 2006; Wang, 2013a), can trigger visibility deterioration and result in haze events (Singh and Dey, 2012; Li et al., 2014). In addition, fine particulate matter with aerodynamic diameters smaller than 2.5 μm ($PM_{2.5}$) can also enter into the alveoli to cause severe cardiovascular diseases, respiratory diseases and even lung cancer (Pope and Dockery, 2006; Gao et al., 2015a). All these impacts have attracted considerable attention among the public and policy makers in East Asia, and the research on aerosols has become a hot topic which is frequently reported and deeply studied during recent years.

In order to better understand the properties of atmospheric aerosols and their impacts, chemical transport models (CTMs) can be a critical tool, and they have been drawn up and applied to study various air pollution issues all over the world. For example, a fully coupled online Weather Research and Forecasting/Chemistry (WRF/Chem) model was developed by Grell et al. (2005) and was used to study the aerosol–radiation–cloud feedbacks on meteorology and air quality (Gao et al., 2014; Zhang et al., 2015a; Qiu et al., 2017); a Models–3 Community Multi–scale Air Quality (CMAQ) modeling system was designed by the US Environmental Protection Agency (Byun and Ching, 1999) and was carried out to address acid deposition, visibility and haze pollution issues (Zhang et al., 2006; Han et al., 2014; Fan et al., 2015); a nested air quality prediction model system (NAQPMS) was developed by the Institute of Atmospheric Physics, Chinese Academy of Science (IAP/CAS) (Wang et al., 2001) for targeting at reproducing the transport and evolution of atmospheric pollutants in Asia (Li et al., 2012a; Wang et al., 2013c; Li et al., 2017a); a global three–dimensional chemical transport model (GEOS–CHEM) was first presented by Bey et al. (2001) and was applied to study the source sector contribution, long–range transport and the prediction of future change in ozone and aerosol concentrations (Liao et al., 2006; Li et al., 2016b; Zhu et al., 2017).

Although significant advances have taken place in these CTMs, how to accurately reproduce and/or predict the concentrations and the distributions of atmospheric pollutants is still a challenge, with the problems of inaccurate emission inventories, poorly represented initial and boundary conditions, and imperfect physical, dynamical and chemical parameterizations (Carmichael et al., 2008). Meanwhile, most CTMs are designed to focus on the air quality over developed countries, such as Europe and America, rather than in Asia, and the assumptions or look–up tables used in models may not be suitable to simulate the Asian environment (Gao et al., 2018). Therefore, before providing scientifically meaningful



information and answering "what–if" questions for policy makers, model performances must be first evaluated. Hayami et al. (2008) and Mann et al. (2014) pointed out that multi–model ensemble mean (EM) tends to show better performance than most participant models when comparing with observations, and large variation in simulation results can be found among participant models, which may be caused by using different parameters and calculation methods in each CTM (Carmichael et al., 2002; Hayami et al., 2008; Wang et al., 2008; Holloway et al., 2008). In order to develop a better common understanding of the performance and uncertainties of CTMs in East Asia applications, and to acquire a more mature comprehension of the properties of atmospheric aerosols and their impacts in East Asia, a model inter–comparison study should be initiated, and Model Inter–Comparison Study for Asia (MICS–Asia) gives an opportunity to investigate these questions. Meanwhile, model inter–comparison study in East Asia is very limited (Phadnis et al., 1998; Kiley et al., 2003; Han et al., 2008) and far more efforts are needed.

The MICS–Asia project was initiated in 1998. In the first phase of MICS–Asia (MICS–Asia Phase I), the primary target is to study the long–range transport and deposition of $SO_4^{2-}$ in East Asia by analyzing the submitted simulation results from eight CTMs. Source–receptor relationship, contributions of wet/dry pathways for remove, and the influence of model structure and parameters on simulation capability are also estimated. More details can be found in Carmichael et al. (2002). As an extension of Phase I, MICS–Asia Phase II includes more chemical species of concern, such as sulfur, nitrogen and ozone. This broader collaborative study examined four different periods, encompassing two different years and three different seasons (March, July, and December in 2001, and March in 2002). Simulation results are from nine different regional modeling groups. Detailed information about this project can be found on the overview paper of Carmichael et al. (2008). In 2010, the MICS–Asia III project was launched. As a part of EANET additional research activity and a continuing research of MICS–Asia series, three topics are discussed, including comparison and evaluation of current multi–scale air quality models (Topic 1), development of reliable emission inventories for CTMs in Asia (Topic 2), and interactions between air quality and climate changes (Topic 3).

This manuscript focuses on the first topic of the MICS–Asia Phase III, and tries to present and summary the following three objectives, which mainly specialize in the analysis topic of aerosol species. Firstly, a comprehensive evaluation of the strengths and weaknesses of current multi–scale air quality models for simulating particulate matter (PM) is provided against extensive measurements from in–situ and satellites, aiming to show the capability of participant models. Secondly, the diversity of simulated aerosol concentrations among participant models is analyzed including suggestions about how to reduce uncertainties in simulation results, which can be used as a reference for future development and improvement of models. Thirdly, characteristics of aerosol chemical components over analyzed regions in East Asia are revealed, which may be helpful to provide confidence for future investigation of aerosol impacts on regional climate in East Asia.

The descriptions of model configurations, model inputs, analyzing area and observation data are presented in Section 2. The evaluation for model performance and the inter–comparison between participant models are shown in Section 3. The conclusions and discussions are presented in Sections 4.



## 2 Inter–comparison framework

### 2.1 Model description

Fourteen modeling groups participated in MICS–Asia phase III Topic 1. Basic information about the configurations of each model is summarized in Table 1. Among these models, five different kinds of chemistry modules are applied, including
CMAQ (Community Multiscale Air Quality), WRF–Chem (Weather Research and Forecasting Model coupled with Chemistry), NAQPMS (nested air quality prediction model system), NHM–Chem (non–hydrostatic mesoscale model coupled with chemistry transport model), and GEOS–Chem (global three–dimensional chemical transport model). For CMAQ, there are four different versions: CMAQ5.0.2 (M1 and M2), CMAQ5.0.1 (M3), CMAQ4.7.1 (M4, M5 and M6), and CMAQ4.6 (M14). For WRF–Chem, there are also four different versions: WRF–Chem3.7.1 (M7), WRF–Chem3.6.1 (M8),
WRF–Chem3.6 (M9), and WRF–Chem3.5.1 (M10). Only one version is used for NAQPMS (M11), NHM–Chem (M12) and GEOS–Chem (version 9.1.3, M13).

The settings of gas phase chemistry and aerosol chemistry are key components of chemical transport models, and can influence the simulation results significantly (Cuchiara et al., 2014). The gas chemistry of SAPRC 99 (the 1999 Statewide Air Pollution Research Center) was used in M1, M2, M4, M5, M6, M12 and M14. It is a detailed mechanism for the
gas–phase atmospheric reactions of volatile organic compounds (VOCs) and oxides of nitrogen ($NO_x$) in urban and regional atmospheres. It includes 76 species reacting in 214 reactions (Carter, 2000). CB05 (2005 Carbon Bond) chemical mechanism was used in M3. It is a condensed mechanism of atmospheric oxidant chemistry that provides a basis for computer modeling studies of ozone, particulate matter (PM), visibility, acid deposition and air toxics issues, with 51 species and 156 reactions (Yarwood et al., 2005). M9 and M10 used RADM2 (Regional Acid Deposition Model, version 2) gas chemistry mechanism.
The inorganic species included in the RADM2 are 14 stable species, 4 reactive intermediates, and 3 abundant stable species. Atmospheric organic chemistry is represented by 26 stable species and 16 peroxy radicals (Stockwell et al., 1990). It has been extensively used in atmospheric models to predict concentrations of oxidants and other air pollutants. Based on RADM2, Regional Atmospheric Chemistry Mechanism (RACM) created by Stockwell et al. (1997) is capable of modeling a wide variety of complex and practical situations, from the Earth's surface to the troposphere and from remote to polluted
urban conditions. This mechanism was used in M7 and M8, including 17 stable inorganic species, 4 inorganic intermediates, 32 stable organic species, and 24 organic intermediates, with up to 237 reactions. M11 used CBMZ (Carbon–Bond Mechanism version Z) and this mechanism extends the original framework of CBM–IV to function properly at larger spatial and longer timescales, with 67 species and 164 reactions (Zaveri and Peters, 1999). In order to have a comprehensive understanding of factors controlling tropospheric ozone, one major theme of the gas chemistry mechanism used in M13
(GEOS–Chem) is the simulation of ozone–$NO_x$–hydrocarbon chemistry, which includes about 80 species and 300 chemical reactions (Bey et al., 2001; Zhu et al., 2017).

The Aero5/Aero6 aerosol module with ISORROPIA aerosol thermodynamics v1.7/v2 were used in M1, M2, M3, M4, M5, M6, M11, M12 and M14. It is designed to simulate the thermodynamic equilibrium of inorganic species (e.g. $NH_4^+$,



sodium, chloride, $NO_3^-$, $SO_4^{2-}$ and water), and all the aerosol particles are assumed to be internally mixed (Nenes et al, 1998). Meanwhile, these modules use a weighted average approach to approximate the aerosol composition in mutual deliquescence regions, which can speed up the solution time. The Aero5 ISORROPIA (v1.7) was mainly used in CMAQ model version before 5.0, after which the second version (ISORROPIA (v2)) was implemented. The main change in ISORROPIA v2 is to introduce thermodynamics of crustal species such as $Ca^{2+}$, $K^+$ and $Mg^{2+}$ (Fountoukis and Nenes, 2007), and the corresponding impacts are mainly on the gas–particle partitioning of $NO_3^-$ and $NH_4^+$ in areas with high dust emissions. Wang et al. (2012b) pointed out that the updated treatment of crustal species in ISORROPIA v2 can reduce fine mode of particle matter over polluted areas. The aerosol module used in M7 and M9 is MADE (Modal Aerosol Dynamics Model for Europe) (Ackermann et al., 1998) for the inorganic fraction, and the Secondary Organic Aerosol Model (SORGAM) (Schell et al., 2001) for the carbonaceous secondary fraction. For MADE/SORGAM, the modal approach with three log–normally distributed modes (nuclei, accumulation and coarse mode) is implemented in the WRF–Chem model. Similar as the aerosol chemistry of MADE/SORGAM, secondary organic aerosols (SOA) was also simulated by the advanced Volatility Basis Set (VBS) approach in M8 (Tuccella et al., 2015). The bulk GOCART (Goddard Global Ozone Chemistry Aerosol Radiation and Transport) aerosol module (originally developed by NASA) used in M10 can output fourteen aerosol species, including hydrophobic and hydrophilic organic carbon (OC1 and OC2) and black carbon (BC1 and BC2), $SO_4^{2-}$, dust in five particle–size bins, and sea salt in four particle–size bins. This mechanism can provide an integrated sectional scheme for dust emission and aerosol advection, but indirect effects and wet scavenging/deposition schemes regarding cloud interactions are not supported (Chin et al., 2000). Aerosol species considered in GEOS–Chem (M13) include $SO_4^{2-}$ (Park et al., 2004), $NO_3^-$ (Pye et al., 2009), $NH_4^+$, OC and BC (Park et al., 2003), mineral dust (Fairlie et al., 2007), and sea salt (Alexander et al., 2005).

As is known to all that meteorological fields has a profound impact on air quality, and aerosol compositions can affect weather and climate directly by changing clouds, radiation, and precipitation (Forkel et al., 2015). In order to simulate the concentrations of air pollutants, meteorological models and chemistry transport models can be implemented either offline or online (Kong et al., 2015). Offline modeling implies that CTM is run after the meteorological simulation is completed, and the chemistry feedbacks on meteorology are not considered. Online modeling allows coupling and integration of some of the physical and chemical components (Baklanov et al., 2014). According to the extent of online coupling, there are two ways of coupling: online integrated coupling (meteorology and chemistry are simulated in one model using the same grid and using one main time step for integration) and online access coupling (meteorology and chemistry are independent, but information can be exchanged between meteorology and chemistry on a regular basis) (Baklanov et al., 2014). Among these participating models, M4, M5, M6, M12, M13 and M14 are offline models. M1, M2, M3, and M11 are online access models. M7, M8, M9 and M10 are online integrated models. Different coupling methods can cause different simulation results due to the interactions among aerosol, weather and climate. Even though using the same coupling way, different parameterizations can also cause uncertainties. More details about the model configurations are summarized in Table 1 and other MICS–Asia Phase III companion papers.



For aerosol species, modeling variables of BC, OC, $SO_4^{2-}$, $NO_3^-$, $NH_4^+$, $PM_{2.5}$, $PM_{10}$ and AOD from the fourteen participant models, as listed in Table 2, are requested to upload to a public server. But no data are acquired from M10, and all simulation results from M3 are incredible. Therefore, only twelve models are analyzed in this manuscript. Meanwhile, M5, M6 and M8 did not submit simulated AOD. M13 did not submit simulated $PM_{10}$. M7 did not submit OC. Neither BC nor OC was submitted from M9.

## 2.2 Information about model inputs

Based on the experience of Phase I and Phase II, all participant models in Phase III Topic 1 are required to use common meteorological fields, emission inventories and boundary conditions in order to reduce the potential diversity that may be caused by input dataset.

The meteorological fields are outputted from the Weather Research and Forecasting Model (WRF v3.4.1) using the National Center for Environmental Prediction (NCEP) Final Analysis (FNL) data with 1 °×1 ° spatial resolution and 6 h temporal interval, but M10, M12, M13 and M14 choose to use others. In M10, the initial and lateral boundary meteorological fields are run by WRF (v3.5.1) driven by Modern Era Retrospective-analysis for Research and Applications (MERRA) reanalysis dataset. The outputs from the Japan Meteorological Agency (JMA) non–hydrostatic mesoscale model (NHM) are initialized in M12 (Kajino et al., 2012). M13 is driven by assimilated meteorological data from the Goddard Earth Observing System (GEOS) of NASA's Global Modeling and Assimilation Office (Chen et al., 2009; Li et al., 2016c). Although initial and lateral boundary conditions are taken from the same NCEP FNL data, three dimensional meteorological fields used in M14 are simulated by Regional Atmospheric Modeling System (RAMS) (Zhang et al., 2002, 2007; Han et al., 2009, 2013). These different atmospheric forcing dataset may result in differences in simulated circulation fields and other meteorological variables, which can further influence the concentrations and the distributions of simulated air pollutants.

All models utilized a common emission inventory, which includes anthropogenic, biogenic, biomass burning, air and ship, and volcano emissions. The anthropogenic emission dataset over Asia, named MIX, is developed by harmonizing five regional and national emission inventories with a mosaic approach. These five inventories are REAS2 (REAS inventory version 2.1 for the whole of Asia, Kurokawa et al., 2013), MEIC (the Multi-resolution Emission Inventory for China developed by Tsinghua University), PKU–NH$_3$ (a high–resolution NH$_3$ emission inventory by Peking University, Huang et al., 2012), ANL–India (an Indian emission inventory developed by Argonne National Laboratory, Lu et al., 2011), and CAPSS (the official Korean emission inventory form the Clean Air Policy Support System, Lee et al., 2011). The MIX inventory includes ten species (SO$_2$, NO$_x$, CO, CO$_2$, NMVOC (non–methane volatile organic compounds), NH$_3$ (ammonia), BC (black carbon), OC (organic carbon), PM$_{2.5}$ and PM$_{10}$) in each sector (power, industry, residential, transportation, and agriculture), and is developed for the year 2010 with monthly temporal resolution and 0.25 degree spatial resolution. More details can be found in Li et al. (2017b). Weekly and diurnal profiles of the anthropogenic emissions provided by the MICS–Asia organizers are used in model simulations, including the emission factors for the first seven vertical levels (Fig. S1). Hourly biogenic emissions quantified by the Model of Emissions of Gases and Aerosols from Nature (MEGAN) version





2.04 (Guenther et al., 2006) are provided for the whole MICS–Asia phase III simulation period. To drive MEGAN, meteorological variables (e.g. solar radiation, air temperature, soil moisture) and land cover information (e.g. leaf area index (LAI), plant functional types (PFTs)) are necessary inputs, and these data are obtained from WRF simulations and MODIS (Moderate Resolution Imaging Spectroradiometer) products, respectively. Biomass burning emissions are processed by

re–gridding the Global Fire Emissions Database (GFED) version 3 (van der Werf et al., 2010). Hourly fraction of biomass burning emission for each day during the entire year is also provided. The aircraft and shipping emissions are based on the 2010 HTAPv2 (Hemispheric Transport of Air Pollution) emission inventory (0.1 by 0.1 degree) (Janssens–Maenhout et al., 2015). Daily volcanic $SO_2$ emissions can be collected from the AEROCOM program (http://www-lscedods.cea.fr/aerocom/AEROCOM\HC/volc/, Diehl et al., 2012; Stuefer et al., 2013). The spatial distribution

of the merged emissions of $SO_2$, $NO_x$ and $PM_{2.5}$ from anthropogenic, biogenic, biomass burning, air and ship, and volcano emissions are shown in Fig. 1. Similar spatial patterns can be found among these species, with high values in eastern China and northern India.

Chemical concentrations at the top and lateral boundary conditions from 3–hourly global CTM outputs of CHASER (run by Nagoya University, Sudo et al., 2002a; Sudo et at., 2002b) and GEOS–Chem (run by University of Tennessee,

http://acmg.seas.harvard.edu/geos/) are provided by MICS-Asia III. CHASER model is run with 2.8 °×2.8 ° horizontal resolution and 32 vertical layers, and GEOS-Chem is run with 2.5 °×2 °horizontal resolution and 47 vertical layers.

A reference model computational domain recommended by MICS–Asia organizers covers the region of (15.4 °S–58.3 °N, 48.5 °E–160.2 °E) using 180×170 grid points at 45 km horizontal resolution, but most participant models employ different extents of the region as shown in Fig. 2. In order to minimize the influence from lateral boundary conditions and to cover

most high–profile areas of East Asia, a common domain is designed in this manuscript as shown in Fig. 2 with the red solid line. For M13 and M14, missing value is used to fill the grids beyond their simulation domains. In this common domain, five different regions are assigned with different colors (Fig. 3): Region_1, filled with blue, contains Korean Peninsula and Japan; Region_2, filled with cyan, contains China only; Region_3, filled with chartreuse, contains Mongolia and parts of Russia; Region_4, filled with orange, contains most countries in Southeast Asia; Region_5, filled with purple, contains most

countries in South Asia. Therefore, modeling results in different geographical sub-regions of East Asia can be analyzed and compared with each other to show the simulation performance of current CTMs.

The whole year 2010 is chosen as the study period for MICS–Asia Phase III Topic 1. During 2010, many important weather events have been documented, such as extreme summer heat waves and widespread monsoon precipitation which affected many Asian countries (Chen et al., 2015; Jongman et al., 2015); a super dust storm originated from Gobi Desert in

March 2010 and swept across vast areas of East Asia (Li et al., 2011); a winter severe haze episode occurred in the North China Plain (NCP) in January 2010 (Gao et al., 2016b). All these provide good opportunities to analyze the characteristics of the spatial and temporal distribution of aerosol concentrations over East Asia.



### 2.3 Observation data

In order to make an international common understanding and improve air pollution modeling in East Asia, observation data (e.g. $SO_4^{2-}$, $NO_3^-$, $NH_4^+$, $PM_{2.5}$ and $PM_{10}$) at 39 sites of the Acid Deposition Monitoring Network in East Asia (EANET) are used to evaluate the model performance, as did in MICS–Asia Phase II. Common quality assurance and quality control

standards promoted by the ADORC (Acid Deposition and Oxidant Research Center) are adopted among these EANET stations to guarantee high quality dataset. More information about the EANET dataset can be found at http://www.eanet.asia/index.html. In addition to the EANET data, monthly measurements of air pollutants (e.g. $SO_2$, $NO_2$, $PM_{2.5}$ and $PM_{10}$) over the Beijing-Tianjin-Hebei (BTH) region (19 sites) and the Pearl River Delta (PRD) region (13 sites) provided by the China National Environmental Monitoring Center (CNEMC) are also used to compare with the simulation

results from participating models.

As is known to all, China has been experiencing air pollutions with high concentrations of fine particles, and recent studies highlight the importance of secondary aerosols in the formation of haze episodes (Liu et al., 2013; Sun et al., 2016; Chen et al., 2018). However, observed aerosol components (e.g. $SO_4^{2-}$, $NO_3^-$ and $NH_4^+$) in inland China are only available at one EANET site (the Hongwen site). In order to make the evaluation of the model performance more credible, observed

monthly/seasonal/yearly BC, $SO_4^{2-}$, $NO_3^-$, $NH_4^+$ and $PM_{2.5}$ concentrations at several Chinese stations (five stations for BC, thirteen stations for $SO_4^{2-}$, $NO_3^-$ and $NH_4^+$, and twenty-two stations for $PM_{2.5}$) are collected from published documents (Chen et al., 2012; Li, 2012b; Liu, 2012; Meng et al., 2012; Shao, 2012; Wang et al., 2012a; Xu, 2012; Xie et al., 2013; Yu, 2013; Zhao et al., 2013; Tao et al., 2014; Wang, 2014a; Li, 2015; Sun et al., 2015; Wang et al., 2015; Zhang, 2015b; Lai et al., 2016; Li et al., 2016a; Wang et al., 2016b; Deng et al., 2016; Yao et al., 2016).

The Aerosol Robotic Network (AERONET), a ground–based remote–sensing aerosol network consisting of worldwide automatic sun– and sky–scanning spectral radiometers (Holben et al., 1998), provides the aerosol optical depth (AOD) products at 440 nm and 675 nm, which are used to calculate the AOD at 550 nm with the Angström exponent. The AERONET Level 2.0 monthly AOD data (cloud–screened and quality–assured data) at thirty–three sites are utilized in this study. Meanwhile, satellite–retrieved 550 nm AOD products from the Moderate Resolution Imaging Spectroradiometer

(MODIS) and Multi–angle Imaging Spectroradiometer (MISR) are also used to compare with the simulations.

Figure 3 shows the locations of all the observational sites (marked with black dots) for each measured species. Most $SO_4^{2-}$, $NO_3^-$ and $NH_4^+$ monitoring sites are located in China, Japan and the Southeast Asia, only two in Mongolia and four in Russia. Except three $PM_{10}$ sites are located in the Southeast Asia, other PM observational stations are in China and Japan. Detailed information about all these ground–level stations can be found in Table S1 and Table S2.

In general, the wide variety of measurements from in–situ and satellites used in this manuscript can allow for a rigorous and comprehensive evaluation of model performances.



## 3 Results and discussions

### 3.1 Model evaluation

Following the objective of MICS−Asia Phase III Topic 1, comparisons of aerosol concentrations (BC, $SO_4^{2-}$, $NO_3^-$, $NH_4^+$, $PM_{2.5}$ and $PM_{10}$), including aerosol optical depth (AOD), between observations and simulations (results from

individual models and EM) are presented to evaluate the performance of current multi−scale air quality models in East Asia simulation, as well as to analyze the differences between participant models.

### 3.1.1 Evaluation for aerosol particles

Figure 4 illustrates the observed and simulated ground level annual mean concentrations of BC, $SO_4^{2-}$, $NO_3^-$, $NH_4^+$, $PM_{2.5}$ and $PM_{10}$. EMs, derived from averaging all the available participating models (except M3 and M10) are also presented

to exhibit a composite of model performances. Monitoring sites are categorized into five regions (Region_1, Region_2, Region_3, Region_4 and Region_5) by their geographic locations as listed in Table S1, and are separated by vertical dashed lines in Fig. 4. Normalized mean biases (NMBs) between observations and EMs in each defined region and the whole analyzed domain are calculated.

Analyzing Fig. 4(a), we can find that most models show good skills in simulating the BC concentration and its spatial

distribution, with high values over North China Plain (NCP) and Yangtze River Delta (YRD) regions, and low values over Central West of China. But the NMB for EM is −15.8%. This underestimation may be attributed to the large negative bias from all participant models at site 24 (the Gucheng site). This station is located in the Hebei province, which is an industrial city, where air pollution is serious and BC emission is large (Wang et al., 2016c). Due to the low reactivity of BC in the atmosphere, the high uncertainty of BC in current emission inputs (Hong et al., 2017; Li et al., 2017b) may explain this

underestimation.

For $SO_4^{2-}$, observations are relative low in Region_1 (mean value is 3.8 μg m$^{-3}$), Region_3 (mean value is 2.5 μg m$^{-3}$) and Region_4 (mean value is 3.5 μg m$^{-3}$), and most models perform well over these regions. But nearly all observed annual mean $SO_4^{2-}$ concentrations in Region_2 are larger than 10 μg m$^{-3}$ (mean value is 16.9 μg m$^{-3}$), and most models fail to reproduce the high magnitude. A large variance can be found among models, e.g. M14 obviously overpredict ground−level

$SO_4^{2-}$, especially in Region_1, whereas M7 and M9 consistently underpredict $SO_4^{2-}$ at nearly all sites. Huang et al. (2014) and Zheng et al. (2015) pointed out that heterogeneous chemistry on the surface of aerosol can enhance the production of $SO_4^{2-}$, especially under polluted conditions (Li et al., 2018). But the mechanism of the heterogeneous uptake of $SO_2$ on deliquesced aerosols may have not been updated in M7 and M9. The model EM better agrees with measurements of $SO_4^{2-}$ concentration than most participating models, and EM can well reproduce the spatial distribution of $SO_4^{2-}$. However,

underestimation is found in each defined region, especially in Region_2 (NMB=−43.5%) and Region_3 (NMB=−35.3%).

Similar spatial distribution of observed $NO_3^-$ concentrations can also be found in Fig. 4(c). The mean values of observed $NO_3^-$ concentrations in each region are 1.5 μg m$^{-3}$ (Region_1), 13.4 μg m$^{-3}$ (Region_2), 0.6 μg m$^{-3}$ (Region_3) and





1.8 µg m$^{-3}$ (Region_4), respectively. Analyzing the performance of each model, a significant overestimation (underestimation) is simulated by M9 (M7 and M8), especially in Region_2. This may result from the biases of model calculation of heterogeneous reactions (Kim et al., 2014), gas-aerosol phase partitioning (Brunner et al., 2014), and deposition (Shimadera et al., 2014). For example, N$_2$O$_5$ hydrolysis is considered in M9, but not in M7 and M8. Su et al. (2016) pointed out that the

hydrolysis of N$_2$O$_5$ can led up to 21.0% enhancement of NO$_3^-$, especially over polluted regions. This may partly explain the differences of simulated NO$_3^-$ concentrations between M7, M8 and M9. Another major possible reason to explain this extreme underestimation of NO$_3^-$ in M7 and M8 is their incorrect treatments of the NH$_3$ emission inputs. As the main alkaline gas in the atmosphere, NH$_3$ can react with H$_2$SO$_4$ and HNO$_3$, which are produced by the oxidation of SO$_2$ and NO$_x$, to form (NH$_4$)$_2$SO$_4$ and NH$_4$NO$_3$, and makes a significant contribution to the formation of secondary inorganic aerosols (Pan

et al., 2016; Zhang et al., 2018). The low simulated concentrations of NH$_4^+$, as shown in Fig. 4(c) and Fig. 17, can also support this explanation. Although the NMB calculated in Region_All (Region_All means the whole analyzed region) for EM is only −1.1%, EM systematically overpredicts observations in Region_1 (NMB=45.2%) and Region_3 (NMB=38.2%), but underpredicts in Region_2 (NMB=−0.7%) and Region_4 (NMB=−44.9%).

Simulated NH$_4^+$ concentrations are associated with the amounts of SO$_4^{2-}$ and NO$_3^-$, but model predictions can

reproduce the measurements relatively well with NMBs ranging from −7.8% to 32.0%. In general, the calculated NMB in Region_All is 4.0%. However, obviously overestimation (underestimation) is also simulated by M14 (M7 and M8), especially in Region_2.

On average, the observed PM$_{2.5}$ concentration in Region_2 is larger than 50 µg m$^{-3}$, while the mean value is only about 10 µg m$^{-3}$ in Region_1. All participating models can generally capture this spatial distribution pattern. However, significant

underestimation is found at the three remote stations (site 1, 2 and 7) in Region_1 with the NMB of −39.0% for EM. Similar negative bias can also be found in Ikeda et al. (2013), who compared CMAQ (v4.7.1) simulation results against observations from the same remote monitoring stations (Rishiri and Oki) throughout the same year 2010. And Ikeda et al. (2013) pointed that the underestimation of organic aerosols caused the negative bias of simulated PM$_{2.5}$ mass concentration. In Region_2, the NMB for EM is −10.0%.

For PM$_{10}$, the mean observed concentrations in each region are 26.6 µg m$^{-3}$ (Region_1), 114.4 µg m$^{-3}$ (Region_2) and 38.1 µg m$^{-3}$ (Region_4), respectively. Comparing with observations, an underprediction tendency can be found among almost all participating models except M14, which predicts higher concentrations in Region_1, especially at coastal sites, such as site 1(Rishiri), 2(Ochiishi), 4(Sadoseki), 7(Oki) and 14(Cheju). The high−value anomalies along coastal areas simulated by M14 can also be found in Fig. 19, and the positive bias may be caused by the emission and gravitational

settling of sea salt. As Monahan and Muircheartaigh (1980) pointed out that sea salt emissions can be enhanced in the surf zone due to increased number of wave breaking events, and the degree of the enhancement highly depends on the 10 m wind speed used in the whitecap coverage parameterization. Meanwhile, higher wind speed at coastal stations was simulated by M14 (RAMSCMAQ) when comparing with observations from previous related studies (Han et al., 2013; Han et al., 2018). In addition, a gravitational settling mechanism of coarse aerosols from upper to lower layers is added in M14, and the net



effect of this update is an increase in PM$_{10}$ concentrations, especially near coastal areas impacted by sea spray (Nolte et al., 2015). In general, the NMB for EM in Region_All is −31.0%.

Figure 5 and Figure 6 show the seasonality of observed and simulated aerosol particle mass concentrations, including BC, SO$_4^{2-}$, NO$_3^-$, NH$_4^+$, PM$_{2.5}$ and PM$_{10}$. All simulations and observations are grouped into five defined regions as illustrated in Fig. 3, with the modeling results sampled at the corresponding observation sites before averaging together. Individual models are represented by the thin grey lines, with the grey shaded areas indicating their spread. The thick black line is the EM. The red solid line is the observational mean and the dashed red lines represent one standard deviation for each group of stations. The correlation coefficients (Rs) for EMs versus the monthly observations are calculated in each panel, and the normalized mean biases (NMBs) in each season (spring: from March to May; summer: from June to August; autumn: from September to November; winter: January, February and December) for EM are also given.

The measured BC concentrations in Region_2 exhibit an obvious seasonal variation with the minimum (~ 3.5 μg m$^{-3}$) during spring and summer, and the maximum (~ 8 μg m$^{-3}$) during late autumn and winter. All participating models can capture this observed seasonality quite well, and all modeling results are within the standard deviation of the observations, but a large inter−model variation is found, especially in winter when BC concentration is high. Different coupling ways between meteorological and chemical modules, as listed in Table 1, can be used to explain this variation. As Gao et al. (2015b), Briant et al. (2017) and Huang et al. (2018) concluded that the online integrated models can simulate higher BC concentrations than offline models, especially during polluted periods. The correlation coefficient for EM is 0.73.

In each month, the mean−observed PM$_{2.5}$ concentration over Region_2 is larger than that in Region_1. This is because the emissions of primary aerosol and precursors in China are larger than that in Japan and Korean Peninsula (Fig. 1). Nearly all models tend to underpredict the magnitude of PM$_{2.5}$ in Region_1 during the whole simulation period with the range of NMB from −44.3% (in winter) to −22.7% (in summer) for EM. The seasonality of modeling PM$_{2.5}$ concentration in Region_2 is better with the R of 0.69 for EM, comparing with the correlation coefficient (R=0.40) in Region_1. In general, the R for EM in Region_All is 0.83 and the NMB ranges from −2.2% to 13.9% among four seasons.

The characteristics of the observed PM$_{10}$ concentrations in Region_1, Region_2 and Region_4 are similar, with the maximum in March and November, and the minimum during summer. M14 consistently overestimates the PM$_{10}$ concentrations in Region_1 for all periods, while others fall within the standard deviation of the observations. The simulated PM$_{10}$ concentrations in Region_2 show less diversity, but nearly all models peak 2 months later. A distinctive seasonality can be found in Region_4 with the maximum (nearly 80 μg m$^{-3}$) in March, but most models cannot reproduce the maximum. This is because the GFED substantially underestimated the biomass burning emissions over Southeast Asia (Fu et al., 2012), especially during March−April when most intense biomass burning occurred in Myanmar, Thailand and other Southeast Asian countries (Huang et al., 2012), and the emission bias is mainly due to the lack of agricultural fires (Nam et al., 2010). Finally, a weak PM$_{10}$ seasonality was simulated by EM with R of 0.58 in Region_4. In Region_all, although consistently underestimation is found during the whole simulation period with NMB ranging from −40.8% to −25.2% for EM, the seasonal cycle can be well captured by EM with R of 0.78.



For $SO_4^{2-}$, $NO_3^-$ and $NH_4^+$ in Region_1, the seasonal variation characteristics of observations are not obvious, with the annual mean values of ~ 4 µg m$^{-3}$ for $SO_4^{2-}$, 1.5 µg m$^{-3}$ for $NO_3^-$ and 1 µg m$^{-3}$ for $NH_4^+$, respectively. A large inter–model spread of simulated $SO_4^{2-}$ is shown in Fig. 6(a1) with the maximum in June. Double–peak curve is displayed in Fig. 6(b1) with the maximums in May and November, and most models significantly overpredict the $NO_3^-$ concentration, especially in summer. Unlike $SO_4^{2-}$ and $NO_3^-$, the simulated monthly $NH_4^+$ concentrations from most models are within the standard deviation of observations, and the R for multi–model mean is highest with the value of 0.74. In Region_2, the observed monthly mean aerosol components are only available at one EANET site (the Hongwen site, located in the eastern coastal area of China), and the seasonality of observed $SO_4^{2-}$, $NO_3^-$ and $NH_4^+$ from this station is obvious with the maximum in spring and winter, and the minimum in later summer and early autumn. Nearly all models tend to underpredict these concentrations, but the EM captures the seasonal cycles relative well with Rs of 0.57 for $SO_4^{2-}$, 0.85 for $NO_3^-$ and 0.86 for $NH_4^+$, respectively. In Region_3, the observed maximum concentrations of $SO_4^{2-}$ and $NH_4^+$ are in winter, but most models cannot reproduce the increasing tendency in the late autumn and the early winter, and then fail to capture the seasonality (Rs of 0.20 for $SO_4^{2-}$, 0.34 for $NO_3^-$ and 0.18 for $NH_4^+$, respectively). This may due to the low emissions of primary aerosol and precursors in Region_3, as shown in Fig.1. In Region_4, the simulated concentrations of $SO_4^{2-}$, $NO_3^-$ and $NH_4^+$ are fairly good when compared with the measurements. The Rs of EM are 0.73 for $SO_4^{2-}$, 0.63 for $NO_3^-$ and 0.73 for $NH_4^+$. Meanwhile, the model diversities are small. Generally, in Region_All, EM can reproduce the magnitudes of observed $SO_4^{2-}$, $NO_3^-$ and $NH_4^+$ fairly well during the whole simulation period, as well as the seasonal variation characteristics.

As mentioned above that observed monthly mean concentrations of aerosol compositions in China are only available at one EANET station (site 17, the Hongwen station) with missing values in June and October. In order to make the evaluation of simulated aerosol chemical components over China more comprehensive, observed seasonal mean concentrations of $SO_4^{2-}$, $NO_3^-$ and $NH_4^+$ collected from published documents are also used to compare with simulations as shown in Fig. 7. M2 and M14 show the reasonable $SO_4^{2-}$ concentrations in the four seasons, while others fail to reproduce the high observed $SO_4^{2-}$ concentrations, with the NMBs ranging from −79.4% (M7) to −28.0% (M12). Most models overestimate the concentrations of $NO_3^-$ and $NH_4^+$ in China, but significant underestimation can be found in M7 and M8 (NMBs are larger than −70%). The underestimation may be due to their incorrect treatments of the $NH_3$ emission inputs, including missing aqueous-phase and heterogeneous chemistry reactions or the implementations of a different gas phase oxidation mechanism (RACM gas phase chemistry mechanism). In fact, the underestimation of $SO_4^{2-}$ and the overestimation of $NO_3^-$ may be the common phenomenon in most current air quality models (Wang et al., 2013b; Gao et al., 2014; Huang et al., 2014; Zheng et al., 2015), and some hypotheses should be deeply tested in future to reduce the deviation, such as (1) missing oxidation mechanism of $SO_2$ may lead to low concentration of $SO_4^{2-}$, which allows for excess $NO_3^-$ in the presence of ammonia, (2) there is an issue with $NO_x$ partitioning or missing $NO_x$ sink. Analyzing the results from ensemble mean, EM shows better performance than participating models, with NMBs of −46.0% for $SO_4^{2-}$, 1.9% for $NO_3^-$ and 13.1% for $NH_4^+$.

Seinfeld and Pandis (2016) pointed out that chemical productions of $SO_4^{2-}$ and $NO_3^-$ are mainly from the gas−phase or



liquid−phase oxidation of $SO_2$ and $NO_2$. Therefore, further comparisons of observed and simulated seasonal cycle of $SO_2$ and $NO_2$ in Region_2 and annual mean concentrations of $SO_2$ and $NO_2$ at corresponding stations are shown in Fig. S2 and Fig. S3, respectively. From Fig. S2, participating models can generally reproduce the seasonality of the two gases, with Rs of 0.61 for $SO_2$ and 0.65 for $NO_2$, respectively. But overestimations (underestimations) of $SO_2$ ($NO_2$) are found in most

simulation periods, not only in China, but also in other defined regions (Fig. S3), and the overestimation (underestimation) of $SO_2$ ($NO_2$) can be used to explain the underestimation (overestimation) of simulated $SO_4^{2-}$ ($NO_3^-$).

### 3.1.2 Evaluation for aerosol optical depth

The seasonal cycle of simulated aerosol optical depth (AOD) at 550 nm is also compared with measurements at thirty–three AERONET stations. Only nine participating models (M1, M2, M4, M7, M9, M11, M12, M13 and M14)

submitted their simulated AOD values, and the EM is calculated by these nine models. From Fig. 8 we can find that most models tend to overpredict AOD during the whole simulation period in Region_1, Region_2 and Region_3 with NMBs of 74.0%, 38.8% and 107.0% for EM, respectively. In Region_4, an obvious seasonality is observed with the maximum in spring and the minimum in summer. Models can capture the seasonality well, although underestimation is found in spring. The R for EM is 0.65 and the NMB is −8.7% in Region_4. Model bias in Region_5 is smaller with the NMB of −4.2% for

EM, but a quite weak seasonality is simulated with underestimation in spring and summer, and overestimation in autumn and winter. Generally, simulated AOD values lie within a standard deviation of the observations in Region_All with a slight overestimation in autumn and winter. The EM can reproduce the seasonal cycle with R of 0.68, and the NMB for EM is 18.7%.

Figure 9 presents the spatial distribution of 550 nm AOD retrieved by MODIS and simulated by the nine models. In this

study, MODIS AOD is collected by the Terra and Aqua satellites during the whole year of 2010. AOD observed from AERONET stations are also shown. In order to quantify the ability of each model to reproduce the spatial distribution of aerosol particles, spatial correlation coefficients are given in the bottom left corner of each panel. Analyzing the observations from MODIS, we can conclude that AOD values are higher in central and eastern China including Sichuan province with the maximum over 1.0. High values can also be found in the north India. Due to dust events happened in spring, AOD values

over the Taklimakan area are also large (~0.5). Comparing with MODIS AOD, almost all models can reproduce the spatial distribution feature with high values in China and India and low values in other countries. The Rs range from 0.78 to 0.86. The model EM captures the AOD spatial variability better with R of 0.87.

Figure 10 shows the differences between model results and MODIS AOD to further discuss the performance of participant models. We can conclude that most models tend to underestimate the AOD values in the eastern coastal regions of

China and the north regions of India where the emissions are large, in addition to the Taklimakan area in China where dust particles can be lifted up frequently. Meanwhile, overestimation is simulated by M2 and M14, especially in the Sichuan province of China. Generally, mean biases averaged over the whole analyzed region for the nine models ranges from −0.16 to 0.05, and the mean bias for EM is −0.08.



Figure 11 shows the annual mean 550 nm AOD from each available model averaged over the five defined regions (Region_1 to Region_5) and the whole analyzed domain (Region_All), together with the measured AOD retrieved from MODIS and MISR. AOD averaged over the corresponding AEROENT stations in each region are also shown. Analyzing the observations, MODIS AOD is the highest and AERONET value is the lowest. This difference can be explained by the

systematic biases in MODIS retrievals due to the impacts of aerosol model assumptions and cloud contamination (Hauser et al., 2005; Toth et al., 2013), in addition to the difference in number of days used to calculate the average (Li et al., 2009). Meanwhile, observations from AERONET sites only represent special samples in each region. Similar results can also be found in other researches (Alpert et al., 2012; Li et al., 2014; Liu et al., 2014). Analyzing the simulations, multi−model mean can generally reproduce the magnitude of observations within a factor from 0.5 to 1.6, especially comparing with AOD

values retrieved from MISR, and the EMs in the five defined regions are 0.29±0.12 for Region_1, 0.34±0.26 for Region_2, 0.21±0.09 for Region_3, 0.18±0.17 for Region_4 and 0.23±0.13 for Region_5, respectively. But the inter−model spread is large by a factor of 2–5 in magnitude, and in most regions, AOD values simulated by M4 are lowest, M2 and M11 show the highest.

### 3.1.3 Statistics for aerosol particles and aerosol optical depth

Table 3 shows the statistics of correlation coefficient (R), normalized mean bias (NMB) and root−mean squared error (RMSE) for BC, $SO_4^{2-}$, $NO_3^-$, $NH_4^+$, $PM_{2.5}$, $PM_{10}$ and AOD. Results from twelve models and EM are compared with available observations. Best results are set to be bold with underline.

It can be found that all models are able to generally capture the variability of BC in China, with Rs ranging from 0.65 of M5 to 0.80 of M8, but nearly all models tend to underestimate the BC concentration, except M1 and M2. The maximum

negative deviation is simulated by M5 with NMB of −54.9%, while the maximum positive deviation is from M2 with NMB of 12.7%. All the RMSEs are less than the mean BC observation (5.0 μg m$^{-3}$). Comparing to the observed $SO_4^{2-}$, most models fail to reproduce the magnitude of concentrations. NMBs range from −67.7% of M7 to 69.3% of M14, and the NMB for EM is −19.1%, meaning underprediction is found in most participating models. This may be caused by the imperfect mechanism of gas−phase and liquid−phase oxidation of $SO_2$, in addition to the missing heterogeneous reactions on the

surface of aerosol particles in most current multi−scale air quality models (Huang et al., 2014, Zheng et al., 2015; Fu et al., 2016). But most models can capture the variation of $SO_4^{2-}$ with Rs ranging from 0.46 of M14 to 0.76 of M13. For $NO_3^-$, Rs vary from 0.29 of M8 to as high as 0.65 of EM. M5 exhibits the largest correlation (0.65) and the smallest NMB (−1.7%) along all models. Although a high R (0.64) is calculated by M9, the NMB is the largest (125.7%). All RMSEs are larger than the measured $NO_3^-$ (1.7 μg m$^{-3}$), meaning a relative poor performance for current air quality models to simulate the $NO_3^-$

concentration in East Asia. For $NH_4^+$, underestimation can be found in M4, M7 and M8, while the others tend to overestimate the $NH_4^+$ concentration. Although all RMSEs are larger than the observed $NH_4^+$ concentration of 1.1 μg m$^{-3}$, most models can capture the variability, with Rs ranging from 0.34 of M8 to 0.75 of M9. Generally, the multi−model mean matches the observed values with R of 0.71, NMB of 14.0% and RMSE of 1.11 μg m$^{-3}$, respectively. Although significant





underpredictions are found in PM$_{10}$ (NMBs range from −55.7% of M5 to −16.9% of M9, except M14), and the inter–model spread of PM$_{2.5}$ is large (NMBs range from −26.5% of M13 to 46.0% of M14), simulated PM$_{2.5}$ and PM$_{10}$ variations are well correlated with measurements (Rs > 0.60), and the RMSEs are all smaller than the averaged measurements (51.4 μg m$^{-3}$ for PM$_{2.5}$ and 80.7 μg m$^{-3}$ for PM$_{10}$, respectively). For AOD, large positive deviation can be found in M2, M9, M11, M13 and

M14, although their Rs are all larger than 0.5. M4 and M7 show the large negative deviation with NMBs of −28.5% and −21.8%, respectively. But their RMSEs are relative small (0.16 for M4 and 0.18 for M7). Generally, the R, NMB and RMSE for EM are 0.68, 18.7% and 0.14, respectively.

### 3.2 Inter–comparison between MICS–Asia Phase II and Phase III

The main purpose of MICS–Asia Phase III Topic 1 is to assess the ability of current multi–scale air quality models to
reproduce air pollutant concentrations. In order to reflect how well the performance of air quality models in East Asia simulation after undergoing substantial development during last several years, statistics (e.g. RMSE and R) for observed and simulated SO$_4^{2-}$, NO$_3^-$ and NH$_4^+$ from MICS–Asia Phase II and Phase III are compared in Fig. 12.

The statistics of MICS–Asia Phase II are taken from Hayami et al. (2008), in which observed monthly mean aerosol composition concentrations were monitored with high completeness at fourteen EANET stations in March, July and
December 2001 and March 2002, while model–predicted monthly surface concentrations are from eight regional CTMs. Notably, NO$_3^-$ and NH$_4^+$ used in Hayami et al. (2008) are total NO$_3^-$ (= gaseous HNO$_3$ + particulate NO$_3^-$) and total NH$_4^+$ (= gaseous NH$_3$ + particulate NH$_4^+$), respectively. More detailed information can be found in Hayami et al. (2008).

Analyzing the RMSEs in Fig. 12, we can conclude that the medians (interquartile ranges) for SO$_4^{2-}$, NO$_3^-$ and NH$_4^+$ are 3.60 μg m$^{-3}$ (3.24, 4.01 μg m$^{-3}$ 25$^{th}$/75$^{th}$ percentiles), 2.76 μg m$^{-3}$ (2.49, 2.96 μg m$^{-3}$ 25$^{th}$/75$^{th}$ percentiles) and 1.28 μg m$^{-3}$
(1.21, 1.47 μg m$^{-3}$ 25$^{th}$/75$^{th}$ percentiles) in Phase III, respectively. Although the medians (except NH$_4^+$) are a little bit larger than that in Phase II, the ranges are quite smaller, meaning similar aerosol concentrations can be simulated by current multi-scale models. Meanwhile, the medians of the correlations of SO$_4^{2-}$, NO$_3^-$, and NH$_4^+$ in Phase III, including the upper and lower quartiles, are all significantly larger than that in Phase II, meaning the better performance of current air quality models in reproducing the variation tendency of observations.

Although the participating models (8 verses 12 CTMs), evaluation sites (14 verses 31 EANET stations) and simulation periods (4 months verses 1 year) are different between Phase II and Phase III, the compared results of statistics calculated from observations and simulations can still generally show that better performance is found in current multi–scale air quality models than those participating in MICS–Asia Phase II when reproducing the concentrations of aerosol particles and their variety characteristics.

### 3.3 Inter–comparison between participant models

Figure 13 to Figure 19 show the spatial distribution of simulated BC, OC, SO$_4^{2-}$, NO$_3^-$, NH$_4^+$, PM$_{2.5}$ and PM$_{10}$



concentrations from each participating model and the multi–model EM. The coefficient of variation (hereinafter, CV), defined as the standard deviation of the models divided by their mean, is also calculated. The larger CV, the lower the consistency is among participating models.

For BC, high values (> 5 µg m$^{-3}$) can be successfully simulated by all models (except M5) over the eastern China including the Sichuan province, and the northeast part of India. Meanwhile, areas with high concentrations (> 5 µg m$^{-3}$) are nearly consistent with the regions where CV values are relative low (< 0.5). However, large CV (> 1.0) is shown over the Himalayas and the Indian Ocean. This is probably due to the different vertical resolutions in addition to the different transmission mechanisms. Generally, the CVs in the five defined regions are all smaller than 0.6. All participant models show similar spatial distribution and magnitude of OC, except M5 and M8 with obvious low values over China and India. Analyzing the results from EM, the highest concentrations are simulated over the Eastern China, Sichuan Province and the northern part of India with values larger than 10 µg m$^{-3}$. CVs are lower than 0.7 in these relative high–concentration areas, while high CV values (> 1.5) are shown over the Tibetan Plateau and low–latitude oceans. For $SO_4^{2-}$, $NO_3^{-}$ and $NH_4^{+}$, high concentrations are centered in the eastern China and the north India, including the Sichuan province of China. However, apparent low $SO_4^{2-}$ ($NO_3^{-}$ and $NH_4^{+}$) concentrations are simulated by M7 and M9 (M7 and M8). Meanwhile, noticeable high concentrations of $SO_4^{2-}$ are simulated by M14, especially along coastal regions. CVs of $SO_4^{2-}$ and $NH_4^{+}$ averaged over the five defined regions are all lower than 0.7, and maximum CVs are all smaller than 1.5, indicating simulation results are in good agreement. But a poor consistency is shown among simulated $NO_3^{-}$ concentrations with CVs larger than 0.6 over each defined region. For PM$_{2.5}$ and PM$_{10}$, high values are simulated by M9, M12 and M14 over arid and semi–arid regions, such as the Taklimakan Desert and the Gobi Desert, where dust events were observed in spring. The CVs in these regions are quite large (over 1.5), which means different processing capacities for dust aerosols and different dust emission mechanisms used among these models. M14 also shows higher values of PM$_{10}$ over coastal regions than other models. This may be caused by the inadequate simulation results of sea salt.

### 3.4 Characteristics of chemical compositions of particulate matter

Figure 20 shows the chemical compositions of simulated particulate matter (PM) averaged over the whole analyzed area in 2010 from each participating model and the multi–model EM. PM$_{10}$ includes PM$_{2.5}$ and OTHER2, while PM$_{2.5}$ is composed of BC, OC, $SO_4^{2-}$, $NO_3^{-}$, $NH_4^{+}$ and OTHER1. Notably, OTHER2 cannot be calculated in M13 because PM$_{10}$ has not been submitted. OC is not available in M7, so we leave it into OTHER1. BC and OC are not available in M9 and these concentrations are grouped into OTHER1.

From Fig. 20 we can find that the simulated concentrations of PM$_{10}$ vary a lot by about a factor of 4 among models, with the highest in M9 (46.5 µg m$^{-3}$) and the lowest in M5 (11.5 µg m$^{-3}$). This large spread can be explained by the differences in simulated concentrations of OTHER2, which is mainly composed of dust aerosol and sea salt aerosol. Generally, the mean PM$_{10}$ concentration from EM is 24.1 µg m$^{-3}$, including 0.9 µg m$^{-3}$ (3.5%) for BC, 2.5 µg m$^{-3}$ (10.3%) for OC, 3.1 µg m$^{-3}$ (12.9%) for $SO_4^{2-}$, 2.7 µg m$^{-3}$ (11.3%) for $NO_3^{-}$, 1.7 µg m$^{-3}$ (7.1%) for $NH_4^{+}$, 6.4 µg m$^{-3}$ (26.7%) for



OTHER1 and 6.8 µg m$^{-3}$ (28.2%) for OTHER2. For PM$_{2.5}$, the regional mean concentration from EM is 17.3 µg m$^{-3}$, with an inter–model range from 9.7 µg m$^{-3}$ of M5 to 28.1 µg m$^{-3}$ of M14. Except OTHER1, the major compositions in PM$_{2.5}$ in East Asia are SO$_4^{2-}$ (18.0%), NO$_3^-$ (15.7%) and OC (14.4%).

Aerosol chemical compositions in six high–profile cities in East Asia (Beijing, Shanghai, Guangzhou, Delhi, Seoul and Tokyo) simulated by each participating model and the multi–model EM are shown in Fig. 21. High values of PM$_{2.5}$ and PM$_{10}$ in Beijing, Shanghai, Guangzhou and Delhi can be simulated by nearly all models, while relative small concentrations are presented in Seoul and Tokyo. For each city, a large spread of PM concentrations can be found among models, this is mainly caused by the differences of the simulated concentrations of OTHER1 and OTHER2. In other words, although common emissions are used, different physical–chemical parameterizations can cause large uncertainties in transmission and remove

processes of aerosols, including the emission processes of dust and sea salt. Analyzing the ratios of aerosol compositions to PM (PM$_{10}$ and PM$_{2.5}$) from simulation results of EM in Fig. 22, the sums of the contributions of BC, OC, SO$_4^{2-}$, NO$_3^-$ and NH$_4^+$ in Beijing, Shanghai, Guangzhou and Delhi are all less than those in Seoul and Tokyo. Among these components in PM$_{2.5}$ (Fig. 22(b1–b6)), except OTHER1, NO$_3^-$ is the major component in Beijing (20.7%) and Delhi (23.6%), while SO$_4^{2-}$ is the major one in Guangzhou (22.2%). Similar contributions of SO$_4^{2-}$ and NO$_3^-$ can be found in Shanghai, Seoul and

Tokyo. All these suggest that different air–pollution control plans should be made in different metropolitans. For seasonal variations of PM$_{2.5}$ concentrations (Fig. 22(c1–c6)), the highest values in Beijing (107.6 µg m$^{-3}$), Shanghai (87.5 µg m$^{-3}$), Guangzhou (59.9 µg m$^{-3}$) and Delhi (108.7 µg m$^{-3}$) are all simulated in winter. This can be explained by their high emissions in winter. However, in Tokyo, the highest PM$_{2.5}$ concentration appears in summer (21.8 µg m$^{-3}$) and the lowest is in winter (10.3 µg m$^{-3}$). In Seoul, PM$_{2.5}$ concentrations are comparable during the four seasons.

**4 Conclusion and Discussion**

     As part of the research of the first topic in MICS–Asia Phase III, this manuscript mainly focuses on the analysis topic of aerosol species, and tries to present and summary the following three objectives: (1) provide a comprehensive evaluation of the strengths and weaknesses of current multi-scale air quality models against extensive measurements from in–situ and satellites, (2) analyze the diversity of simulated aerosol concentrations among participant models, and (3) reveal the

characteristics of key aerosol chemical components over high–profile cities in East Asia. Fourteen regional modeling groups participating in Topic 1 are required to simulate aerosol species using common meteorological fields, emission inventories and boundary conditions during the entire year of 2010 in East Asia. Model predictions are compared with each other, and with measurements of BC, OC, SO$_4^{2-}$, NO$_3^-$ NH$_4^+$, PM$_{2.5}$ and PM$_{10}$. Aerosol optical depth is also rigorously evaluated against observations from AERONET, MODIS and MISR. Note that all simulation results from M3 are incredible, and no

data is gained from M10. Meanwhile, M5, M6 and M8 did not submit simulated AOD. M13 did not submit simulated PM$_{10}$. M7 did not submit OC. Neither BC nor OC was submitted from M9.

     Comparisons against monthly observations from EANET and CNEMC demonstrate that all participant models can



reproduce the spatial–temporal evolution of the concentrations of aerosol species, and multi–model EM shows better performance than most models, with Rs ranging from 0.65 ($NO_3^-$) to 0.83 ($PM_{2.5}$) for EM. Differences between simulations and observations can also be found during the analyzing period, such as $SO_4^{2-}$ is underestimated by participant models (except M12 and M14) with NMBs ranging from −67.7% to −1.6%, while most models overestimate the concentrations of $NO_3^-$ and $NH_4^+$, and the NMBs are 4.9% and 14.0% for EM, respectively. These biases may be caused by the imperfect mechanisms of gas–phase or liquid–phase oxidation of $SO_2$ and $NO_2$, including the missing heterogeneous chemistry reactions in most current multi–scale air quality models. Notably, significant underestimations of $NO_3^-$ and $NH_4^+$ in M7 and M8 may be due to their incorrect treatments of the $NH_3$ emission inputs. The inter–model spread of simulated $PM_{2.5}$ is large, with NMBs ranging from –26.5% of M13 to 46.0% of M14, and nearly all models underestimate the $PM_{2.5}$ concentrations in Region_1. Inaccurate aerosol long–range transport from high–concentration source regions (e.g. Region_2) to low–concentration downstream areas (e.g. Region_1) may explain this bias. Underestimations of $PM_{10}$ are also simulated over the whole analyzed regions, and the NMB of EM in Region_All is −32.6% for $PM_{10}$. For AOD, participating models can reasonably reproduce the spatial variability and the seasonal cycle when comparing with observations from AERONET and MODIS. But underestimations are found along the eastern coastal regions of China and the northern regions of India, where anthropogenic emissions are large, in addition to the Taklimakan area where dust particles can be frequently lifted up. Different capacities to process dust particles and different dust schemes used in participating models may cause the bias.

In order to reveal how well the CTMs can reproduce the characteristics of aerosol species in East Asia after undergoing substantial development during recent years, statistics for observed and simulated $SO_4^{2-}$, $NO_3^-$ and $NH_4^+$ from MICS–Asia Phase II and Phase III are compared. Results obviously show that the variation ranges of RMSEs for each species among participating models in Phase III become smaller, meaning similar concentrations are simulated. Meanwhile the median of the correlations, including the upper and lower quartiles, is larger, indicating the evolution characteristics of observations are better simulated. All these demonstrate a more considerable capacity for reproducing aerosol concentrations and their variation tendencies in current air quality models.

The coefficient of variation is frequently used to quantify the inter–model deviation, and a large CV is calculated over the arid and semi–arid regions, where dust events were observed in the spring of 2010. The poor consistency may be associated with the different dust emission mechanisms used in participating models. But in general, simulation results of BC, OC, $SO_4^{2-}$, $NO_3^-$ and $NH_4^+$ are all in good agreement, especially over the relative highly polluted areas, such as the eastern and northeast China, and the northeast part of India.

According to the simulation results from EM, the highest $PM_{2.5}$ concentrations of Beijing (107.6 μg m$^{-3}$), Shanghai (87.5 μg m$^{-3}$), Guangzhou (59.9 μg m$^{-3}$) and Delhi (108.7 μg m$^{-3}$) are shown in winter, mainly due to the high emissions and unfavorable weather conditions in winter. But the highest in Tokyo appears in summer (21.8 μg m$^{-3}$). $PM_{2.5}$ concentrations are comparable during the four seasons in Seoul. Analyzing the ratios of chemical compositions to $PM_{2.5}$ in these cities, $NO_3^-$ is the major component in Beijing (20.7%) and Delhi (23.6%), $SO_4^{2-}$ is the major one in Guangzhou (22.2%), similar contributions of $SO_4^{2-}$ and $NO_3^-$ are calculated in Shanghai, Seoul and Tokyo. All these suggest that different



air–pollution control plans should be made according to the main contaminants in different cities.

MICS–Asia project gives an opportunity to understand the performance of air quality models in East Asia applications. Analyzing the results concluded above, in order to reduce the diversities of simulated aerosol concentrations among participant models, more efforts are needed for future modeling studies. For example, process analysis scheme should be developed and implemented in air quality models, and individual process, such as advection, diffusion, emission, dry deposition, wet scavenging, gas–phase chemistry and cloud chemistry should be isolated to make a quantitative attribution for the cause of the differences between model predictions. Fully understanding of the source–receptor relationship in each process for a given aerosol species can be helpful to revise parameterization schemes for better simulation capability. Meanwhile, more observations should be collected and used in the next MICS–Asia project.

**Author contribution**

LC, YG and MZ conducted the study design. LC, JZ, HL, JL, KH, BG, XW, YL, CL, SI, TN, MK and KY contributed to modeling data. JF, ZW and JK provided the emission data and observation data. YG and JZ helped with data processing. MZ, JF and JZ were involved in the scientific interpretation and discussion. LC prepared the manuscript with contributions from all co-authors.

**Competing interests**

The authors declare that they have no conflict of interest.

**Acknowledgements**

This study was supported by the National Key R&D Programs of China (No. 2017YFC0209803), the National Natural Science Foundation of China (91544221, 91644215), the University Natural Science Research Foundation of Jiangsu Province (18KJB170012), the Startup Foundation for Introducing Talent of NUIST (2018r007) and the Decision-making Consultation Research Foundation of RICEG, NUIST (2018B33). Monthly pollution concentrations at EANET stations can be collected from http://www.eanet.asia. The AERONET Level 2.0 AOD data is downloaded from https://aeronet.gsfc.nasa.gov/. The MODIS and MISR AOD data are available at https://ladsweb.modaps.eosdis.nasa.gov/ and https://eosweb.larc.nasa.gov/, respectively. Simulation results from the fourteen participating models to generate figures and tables in this manuscript have been archived by corresponding authors, and are available at https://pan.baidu.com/s/1IaaCDhrAR-z2tO6yQNz2cg.



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



**Table 1. Model index, model version, parameterization schemes and reference for each participating model**

| Model Index | Model Version | Gas chemistry | Aerosol chemistry | Dry deposition | Wet scavenging | Meteorology | Boundary Conditiy | Online/Offline | References |
|---|---|---|---|---|---|---|---|---|---|
| M1 | WRFCMAQ5.0.2 | SAPRC99 | Aero6 ISORROPIA(v2) | Wesely | Henry's law | Standard[a] | GEOS-Chem | Online access | Fu et al., (2008) |
| M2 | WRFCMAQ5.0.2 | SAPRC99 | Aero6 ISORROPIA(v2) | Wesely | Henry's law | Standard[a] | Default | Online access | Wang et al., (2014b) |
| M3 | WRFCMAQ5.0.1 | CB05 | Aero6 ISORROPIA(v2) | Wesely | Henry's law | Standard[a] | GEOS-Chem | Online access | Lam et al., (2011) |
| M4 | WRFCMAQ4.7.1 | SAPRC99 | Aero5 ISORROPIA(v1.7) | Wesely | Henry's law | Standard[a] | CHASER | Offline | Itahashi et al., (2014) |
| M5 | WRFCMAQ4.7.1 | SAPRC99 | Aero5 ISORROPIA(v1.7) | M3DRY | Henry's law | Standard[a] | CHASER | Offline | Yamaji et al., (2008) |
| M6 | WRFCMAQ4.7.1 | SAPRC99 | Aero5 ISORROPIA(v1.7) | M3DRY | Henry's law | Standard[a] | CHASER | Offline | Nagashima et al., (2017) |
| M7 | WRFChem3.7.1 | RACM | MADE/SORGAM | Wesely | Walcek and Taylor | Standard[a] | Default | Online integrated | Lee et al., (2017) |
| M8 | WRFChem3.6.1 | RACM | MADE/VBS | Wesely | Henry's law | Standard[a] | CHASER | Online integrated | Lin et al., (2014) |
| M9 | WRFChem3.6 | RADM2 | MADE/SORGAM | Wesely | Easter | Standard[a] | GEOS-Chem | Online integrated | Chen et al., (2017) |
| M10 | WRFChem3.5.1 | RADM2 | GOCART | Wesely | Henry's law | WRF/MERRA2 | MOZART/GOCART[b] | Online integrated | – |
| M11 | NAQPMS | CBMZ | Aero5 ISORROPIA(v1.7) | Wesely | Henry's law | Standard[a] | CHASER | Online access | Wang et al., (2008) |
| M12 | NHMChem | SAPRC99 | ISORROPIA(v2)/MADMS | Kajino, Zhang | Kajino, Pleim and Chang | JMA NHM | CHASER | Offline | Kajino et al., (2012) |
| M13 | GEOS-Chem9.1.3 | Bey | Park, Pye | Wesely | Liu | Geos-5 | GEOS-Chem | Offline | Zhu et al., (2017) |
| M14 | RAMSCMAQ4.6 | SAPRC99 | Aero5 ISORROPIA(v1.7) | Wesely | Henry's law | RAMS/NCEP | GEOS-Chem | Offline | Zhang et al., (2002) |

[a]"Standard' represents the reference meteorological field provided by MICS–Asia III project.

[b]Boundary conditions used in M10 are taken from MOZART and GOCART (Chin et al., 2002; Horowitz et al.,2003), which provides results for gaseous pollutants and aerosols, respectively.

SAPRC99: Carter (2000). CB05: Yarwood (2005). Bey: Bey et al., (2001). CBMZ: Zaveri and Peters (1999). RACM: Stockwell et al., (1997). RADM2: Stockwell et al., (1990). Aero5 ISORROPIA (v1.7): Nenes et al., (1998). Aero6 ISORROPIA (v2): Fountoukis and Nenes (2007). Park: Park et al., (2004). Pye: Pye et al., (2009). MADE–VBS: Tuccella et al., (2015). MADE: Ackermann et al., (1998). M3DRY: Pleim et al., (2001). Wesely: Wesely (1989). Kajino: Kajino et al., (2012). Zhang: Zhang et al., (2001). Liu: Liu et al., (2001). Pleim and Chang: Pleim and Chang (1992). Easter: Easter et al., (2004). Walcek and Taylor: Walcek and Taylor (1986).



**Table 2. Aerosol species simulated by each participating model**

| Model Index | BC | OC | $SO_4^{2-}$ | $NO_3^-$ | $NH_4^+$ | $PM_{2.5}$ | $PM_{10}$ | AOD |
|:---:|:---:|:---:|:---:|:---:|:---:|:---:|:---:|:---:|
| M1 | Y | Y | Y | Y | Y | Y | Y | Y |
| M2 | Y | Y | Y | Y | Y | Y | Y | Y |
| M3 | Y | Y | Y | Y | Y | Y | Y | Y |
| M4 | Y | Y | Y | Y | Y | Y | Y | Y |
| M5 | Y | Y | Y | Y | Y | Y | Y | — |
| M6 | Y | Y | Y | Y | Y | Y | Y | — |
| M7 | Y | — | Y | Y | Y | Y | Y | Y |
| M8 | Y | Y | Y | Y | Y | Y | Y | — |
| M9 | — | — | Y | Y | Y | Y | Y | Y |
| M10 | — | — | — | — | — | — | — | — |
| M11 | Y | Y | Y | Y | Y | Y | Y | Y |
| M12 | Y | Y | Y | Y | Y | Y | Y | Y |
| M13 | Y | Y | Y | Y | Y | Y | — | Y |
| M14 | Y | Y | Y | Y | Y | Y | Y | Y |

"Y" means aerosol species is analyzed in this manuscript.



**Table 3.** Performance statistics of BC, $SO_4^{2-}$, $NO_3^-$, $NH_4^+$, $PM_{2.5}$, $PM_{10}$ and AOD. Best results are set to be bold with underline. The mean observation of each species and the number of observations are also given with italic. In this table, observed monthly mean values from EANET, CNEMC and AERONET are used (except BC, the monthly BC concentrations are collected from published documents).

| Species | Statistics | M1 | M2 | M4 | M5 | M6 | M7 | M8 | M9 | M11 | M12 | M13 | M14 | EM |
|---|---|---|---|---|---|---|---|---|---|---|---|---|---|---|
| **BC** ($5.0\,\mu g\,m^{-3}$) (*nstd=5*) | R | 0.70 | 0.73 | 0.71 | 0.65 | 0.70 | 0.73 | **0.80** | – | 0.69 | 0.68 | 0.75 | 0.72 | 0.73 |
| | NMB(%) | **1.0** | 12.7 | −24.7 | −54.9 | −17.8 | −11.7 | −34.2 | – | −17.5 | −2.2 | −26.8 | −11.6 | −17.0 |
| | RMSE | 4.10 | 4.30 | 2.95 | 4.06 | 2.99 | 2.69 | 2.84 | – | 2.91 | 3.52 | 2.80 | **2.64** | 2.77 |
| **$SO_4^{2-}$** ($3.8\,\mu g\,m^{-3}$) (*nstd=31*) | R | 0.69 | 0.71 | 0.64 | 0.58 | 0.66 | 0.48 | 0.53 | 0.65 | 0.55 | 0.50 | **0.76** | 0.46 | 0.69 |
| | NMB(%) | −23.1 | −13.0 | −31.0 | −26.4 | −26.9 | −67.7 | **−1.6** | −67.0 | −34.5 | 23.2 | −31.9 | 69.3 | −19.1 |
| | RMSE | 3.21 | **3.00** | 3.46 | 3.57 | 3.35 | 4.64 | 3.62 | 4.45 | 3.78 | 4.01 | 3.24 | 5.51 | 3.22 |
| **$NO_3^-$** ($1.7\,\mu g\,m^{-3}$) (*nstd=31*) | R | 0.55 | 0.51 | 0.62 | **0.65** | 0.58 | 0.45 | 0.29 | 0.64 | 0.59 | 0.60 | 0.43 | 0.58 | **0.65** |
| | NMB(%) | 9.0 | −7.2 | −42.7 | **−1.7** | −11.8 | −81.2 | −80.6 | 125.7 | 46.5 | 54.0 | 22.7 | 35.4 | 4.9 |
| | RMSE | 2.70 | 2.71 | 2.48 | 2.29 | 2.46 | 3.37 | 3.18 | 4.37 | 2.89 | 2.80 | 2.96 | 2.62 | **2.27** |
| **$NH_4^+$** ($1.1\,\mu g\,m^{-3}$) (*nstd=31*) | R | 0.67 | 0.64 | 0.68 | 0.66 | 0.69 | 0.55 | 0.34 | **0.75** | 0.66 | 0.62 | 0.64 | 0.68 | 0.71 |
| | NMB(%) | 23.2 | 33.7 | −10.6 | **7.4** | 14.6 | −93.5 | −34.2 | 45.3 | 35.0 | 49.9 | 34.9 | 56.3 | 14.0 |
| | RMSE | 1.24 | 1.42 | 1.15 | 1.21 | 1.16 | 1.83 | 1.53 | 1.26 | 1.27 | 1.54 | 1.29 | 1.47 | **1.11** |
| **$PM_{2.5}$** ($51.4\,\mu g\,m^{-3}$) (*nstd=14*) | R | 0.80 | 0.78 | 0.80 | 0.71 | 0.80 | 0.80 | 0.77 | 0.82 | 0.80 | 0.78 | 0.75 | 0.81 | **0.83** |
| | NMB(%) | 10.0 | 13.6 | **−1.3** | −25.3 | −5.8 | −5.7 | −15.3 | 26.2 | 5.2 | 31.4 | −26.5 | 46.0 | 4.4 |
| | RMSE | 27.56 | 34.88 | 23.03 | 28.00 | 21.80 | 23.54 | 24.83 | 28.52 | 22.06 | 34.87 | 27.10 | 35.85 | **21.23** |
| **$PM_{10}$** ($80.7\,\mu g\,m^{-3}$) (*nstd=51*) | R | 0.75 | 0.74 | 0.74 | 0.65 | 0.75 | 0.70 | 0.70 | 0.66 | 0.78 | **0.82** | – | 0.63 | 0.78 |
| | NMB(%) | −40.7 | −38.7 | −35.7 | −55.7 | −46.6 | −43.7 | −43.4 | −16.9 | −25.4 | −18.8 | – | **7.1** | −32.6 |
| | RMSE | 51.31 | 50.88 | 49.10 | 64.55 | 55.31 | 55.07 | 55.11 | 50.67 | 42.91 | **37.28** | – | 47.26 | 45.81 |
| **AOD** (0.2) (*nstd=38*) | R | 0.64 | 0.55 | 0.56 | – | – | 0.54 | – | 0.60 | 0.69 | 0.66 | **0.71** | 0.57 | 0.68 |
| | NMB(%) | **−2.0** | 63.7 | −28.5 | – | – | −21.8 | – | 11.1 | 73.1 | −6.2 | 47.1 | 36.7 | 18.7 |
| | RMSE | 0.15 | 0.22 | 0.16 | – | – | 0.18 | – | 0.19 | 0.22 | **0.13** | 0.25 | 0.22 | 0.14 |



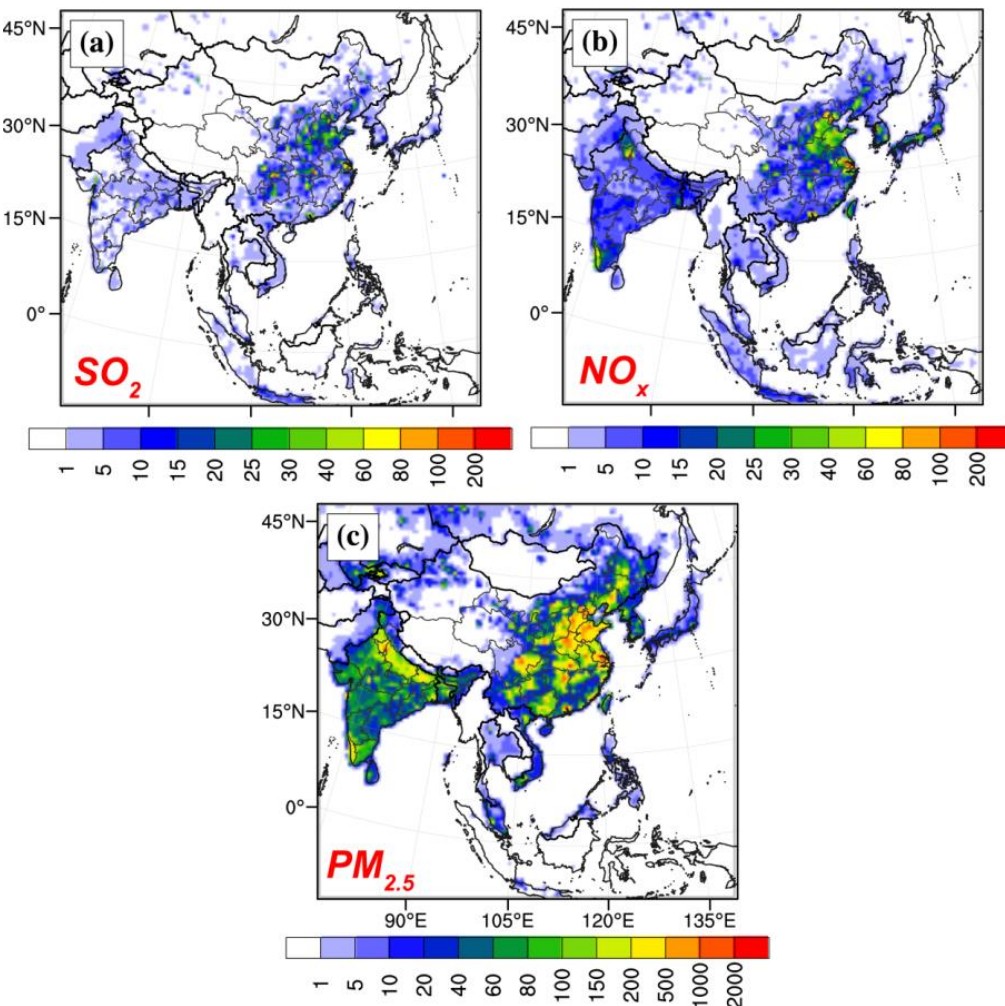

**Figure 1:** The merged emission inventories of MIX emission, MEGAN biogenic emission, GFED biomass burning emission, air and ship emission, and volcanic emission for $SO_2$, $NO_x$ and $PM_{2.5}$ in 2010. The unit for gas is Mmol/month/grid, and the unit for aerosol is Mg/month/grid.





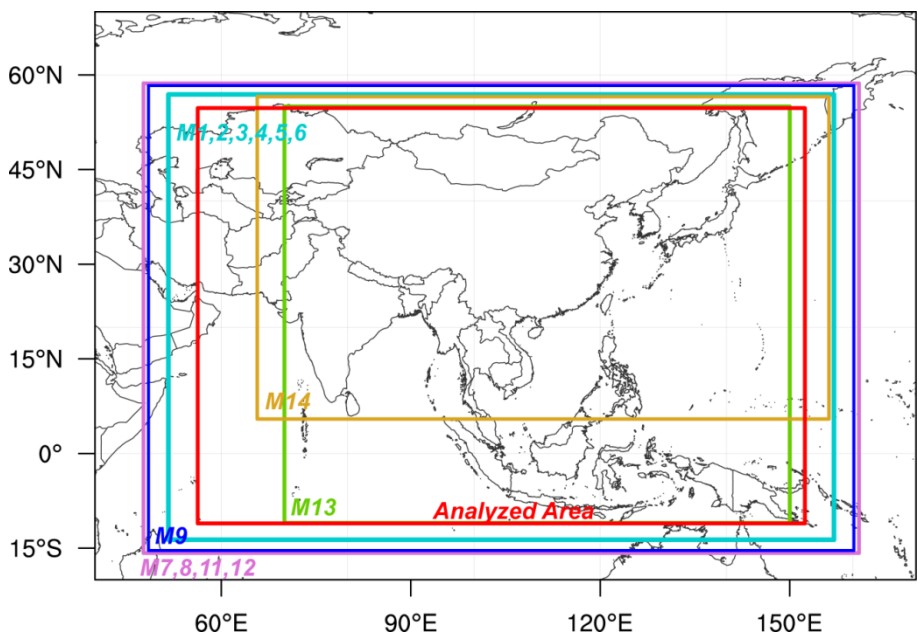

**Figure 2: Simulation domain for each participating model and the final analyzed area used in this manuscript.**





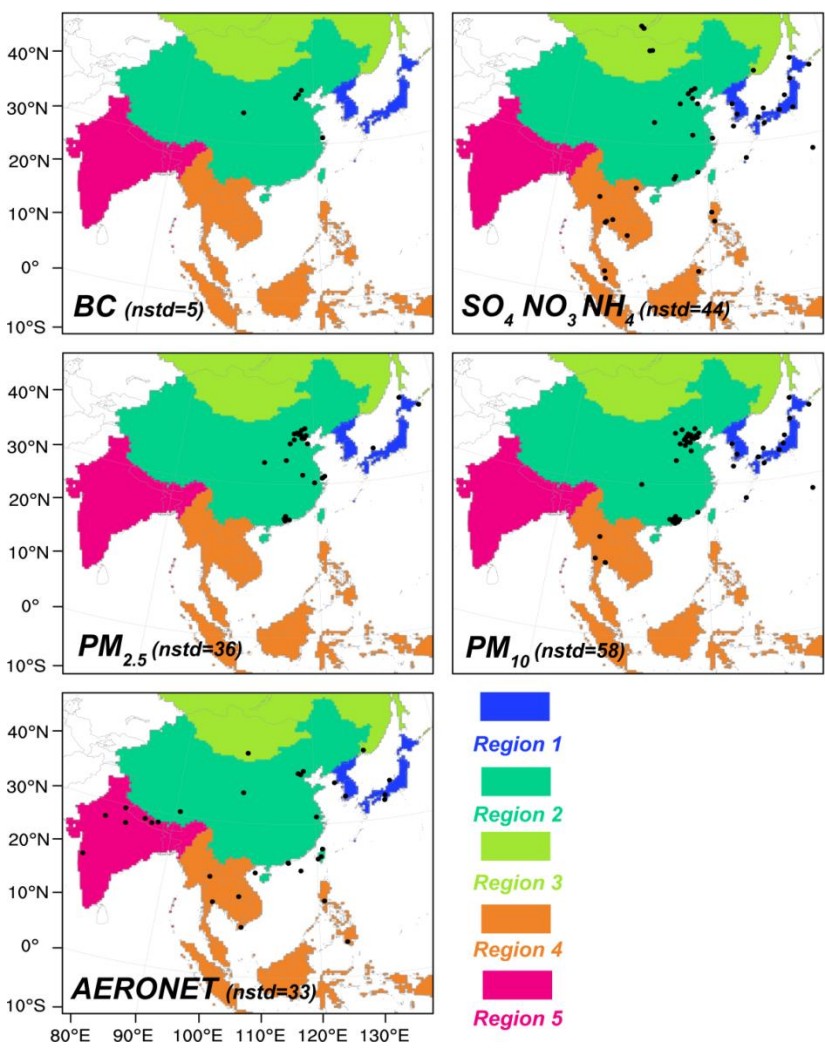

**Figure 3: Spatial distribution of observation sites for each species. Five designed regions are also shown in each panel.**

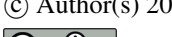



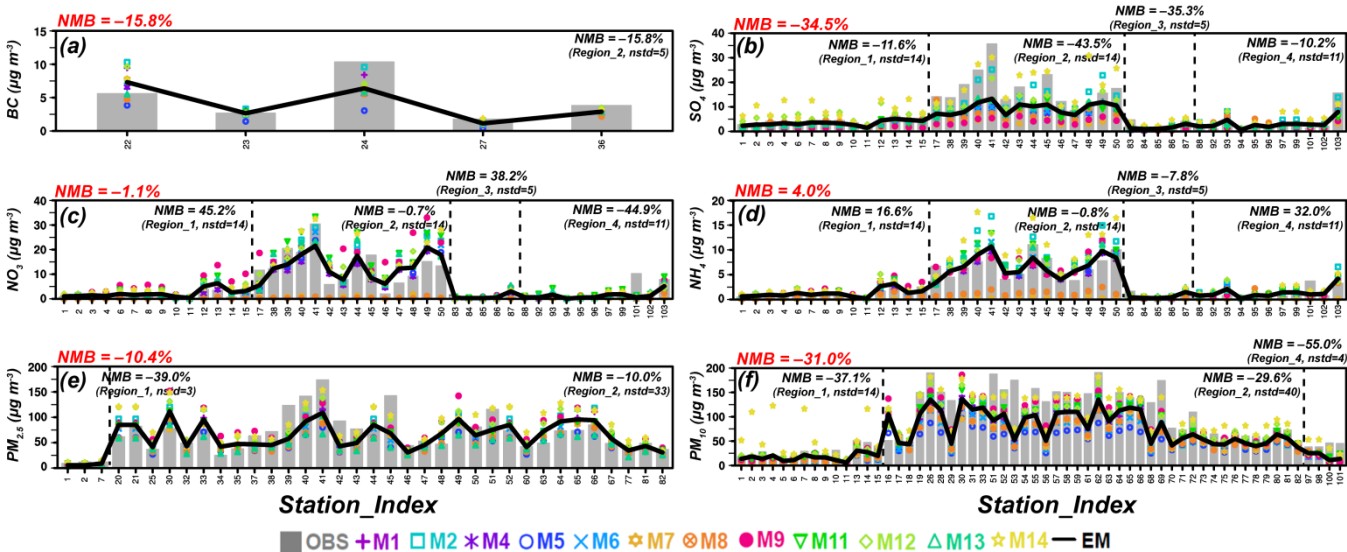

5  **Figure 4: Comparison of observed and simulated concentrations of (a) BC, (b) $SO_4^{2-}$, (c) $NO_3^-$, (d) $NH_4^+$, (e) PM$_{2.5}$ and (f) PM$_{10}$. In each panel, the gray bars show observation data, the colored dots represent simulation results from participating models, and the black solid line is the ensemble mean. The numbers on x-axis represent the monitoring sites, and the information of these sites is listed in Table S1. Normalized mean biases (NMBs) between observations and ensemble means in each defined region (with black color) and the entire analyzed area (with red color) are also shown. In this picture, observed annual mean values from EANET,**

10  **CNEMC and published documents are used.**



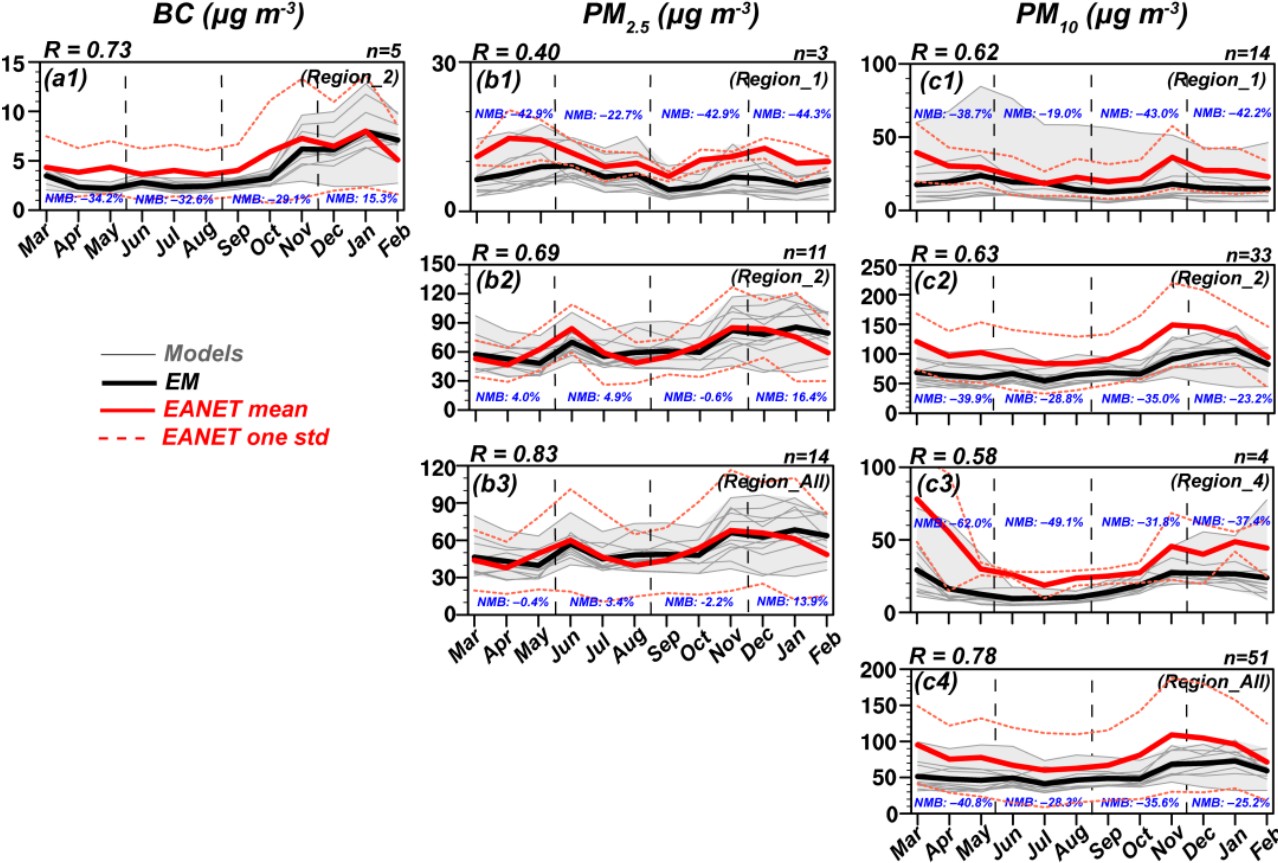

**Figure 5: Observed and simulated seasonal cycle of aerosol species. (a1) BC, (b1)-(b3) PM₂.₅, (c1)-(c4) PM₁₀. Simulations and observations are grouped into five defined regions as illustrated in Figure 3, with each model sampled at the corresponding monitoring sites in each region before averaging together. Individual models are represented by the thin grey lines, with the grey shaded area indicating their spread. The thick black line is the ensemble mean. The red solid line is the observational mean and the dashed red lines mean one standard deviation for each group of stations. The correlations (Rs, with black color) and normalized mean biases (NMBs, with blue color) for ensemble means versus observations during each season (spring: from March to May; summer: from June to August; autumn: from September to November; winter: January, February and December) and the entire year are shown in each panel. Also shown is the number of monitoring sites participating in calculating statistics in each region. In this picture, observed monthly mean values from EANET and CNEMC are used (except BC, the monthly BC concentrations are collected from published documents).**



**Figure 6: Same as Figure 5, but for SO$_4^{2-}$ (a1-a5), NO$_3^-$ (b1-b5) and NH$_4^+$ (c1-c5). In this picture, only monthly EANET observations are used.**





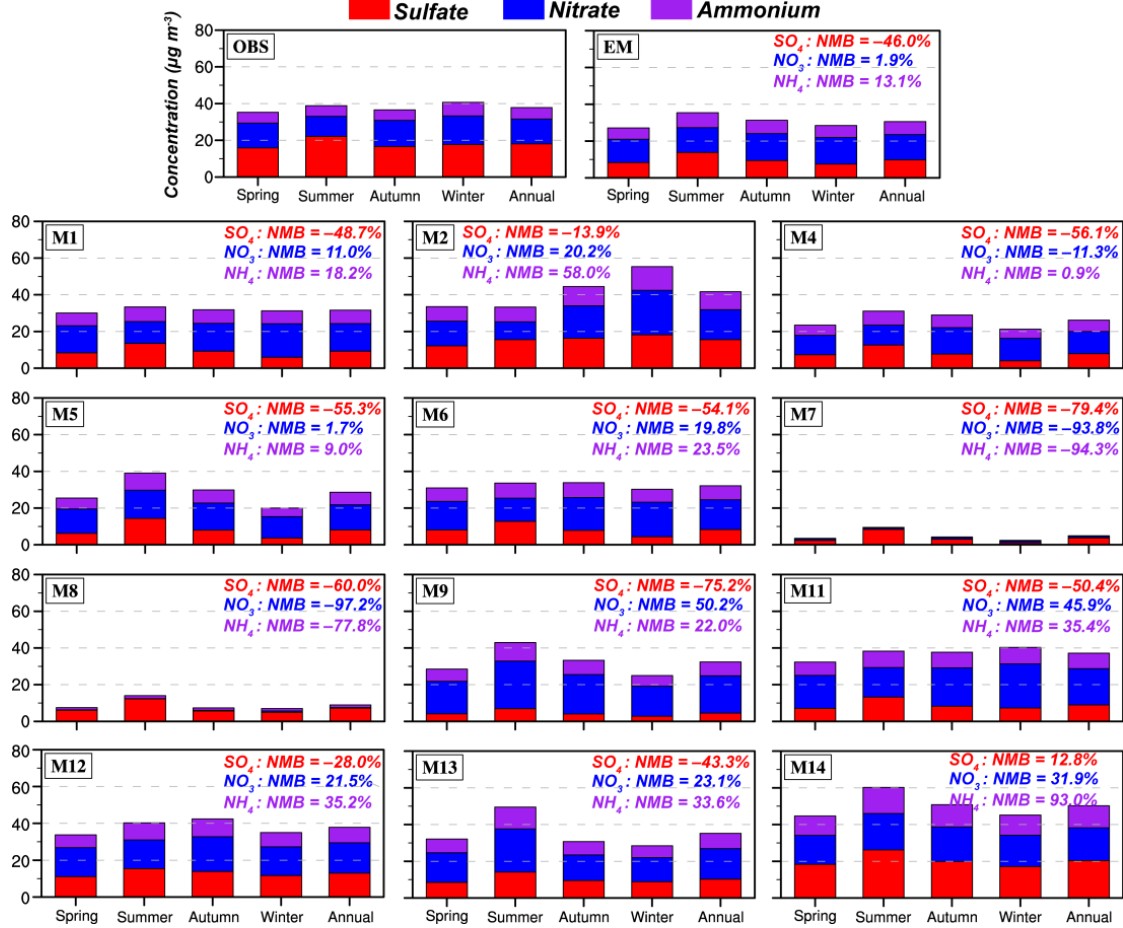

**Figure 7: Observed and simulated seasonal mean concentrations of $SO_4^{2-}$, $NO_3^-$ and $NH_4^+$ in Region_2. Normalized mean biases (NMBs) of $SO_4^{2-}$ (with red color), $NO_3^-$ (with blue color) and $NH_4^+$ (with purple color) for each participating model and the ensemble model are also shown. In this picture, seasonal observations are collected from published documents.**



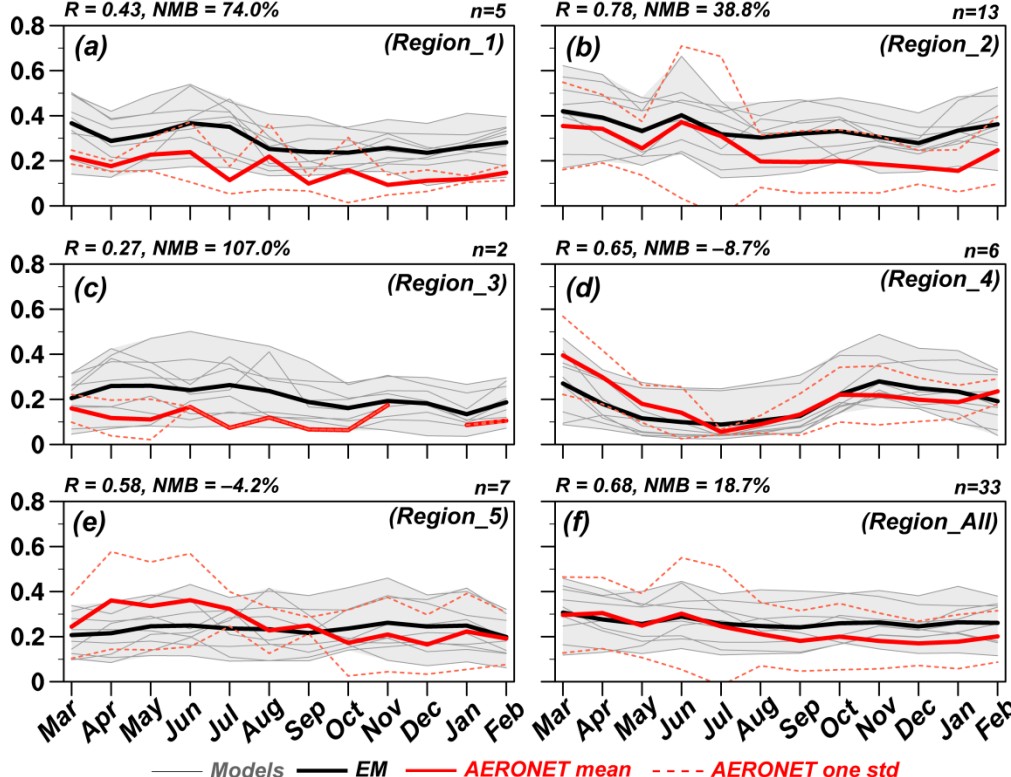

**Figure 8: Similar as Figure 5, but for comparison of seasonal cycle of aerosol optical depth (AOD) at 550 nm between simulations and AERONET observations in each defined region. (a) Region_1, (b) Region_2, (c) Region_3, (d) Region_4, (e) Region_5, and (f) Region_All.**




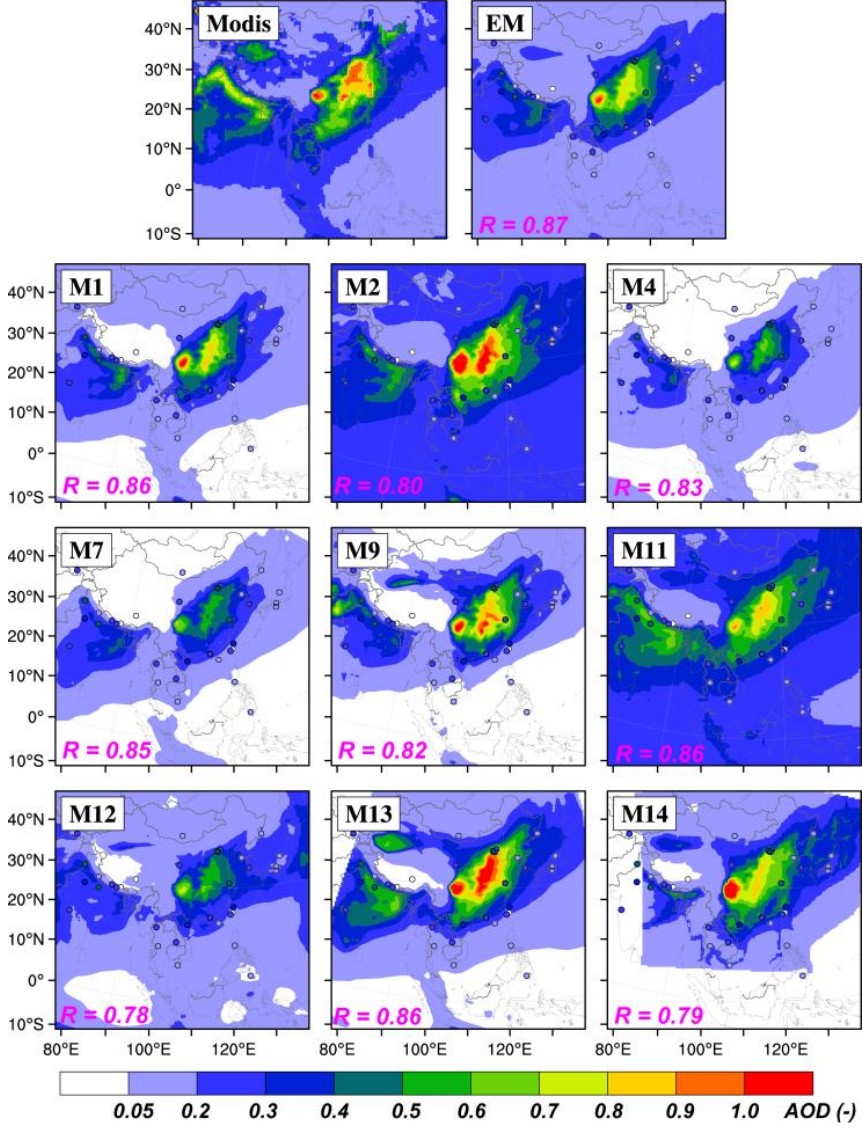

**Figure 9: Spatial distribution of aerosol optical depth (AOD) at 550 nm retrieved by MODIS and simulated by participating models.**
5   **The spatial correlation coefficients are given in the bottom left corner of each panel. Observed AOD from AERONET stations are also shown.**




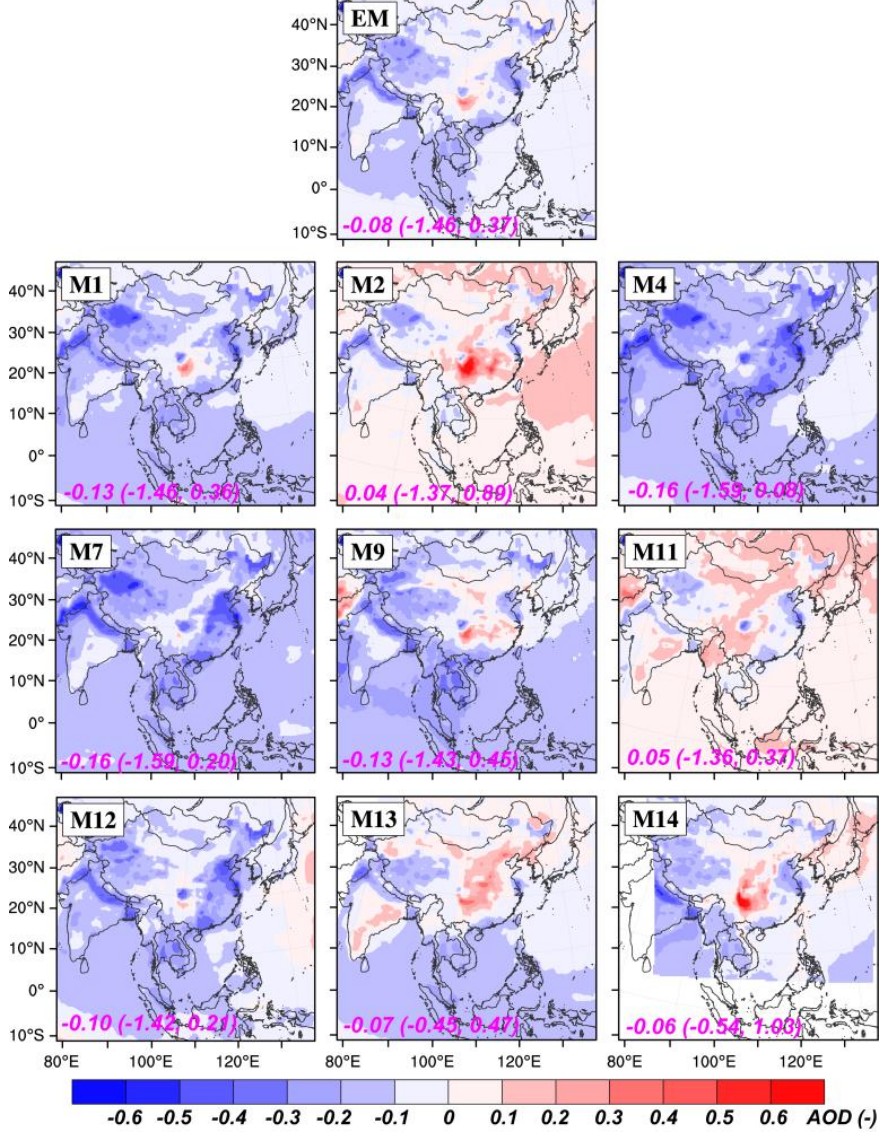

**Figure 10: Spatial distribution of the differences between MODIS AOD and simulation results. The domain mean difference (the minimum difference, the maximum difference) are also listed in the bottom left corner of each panel.**





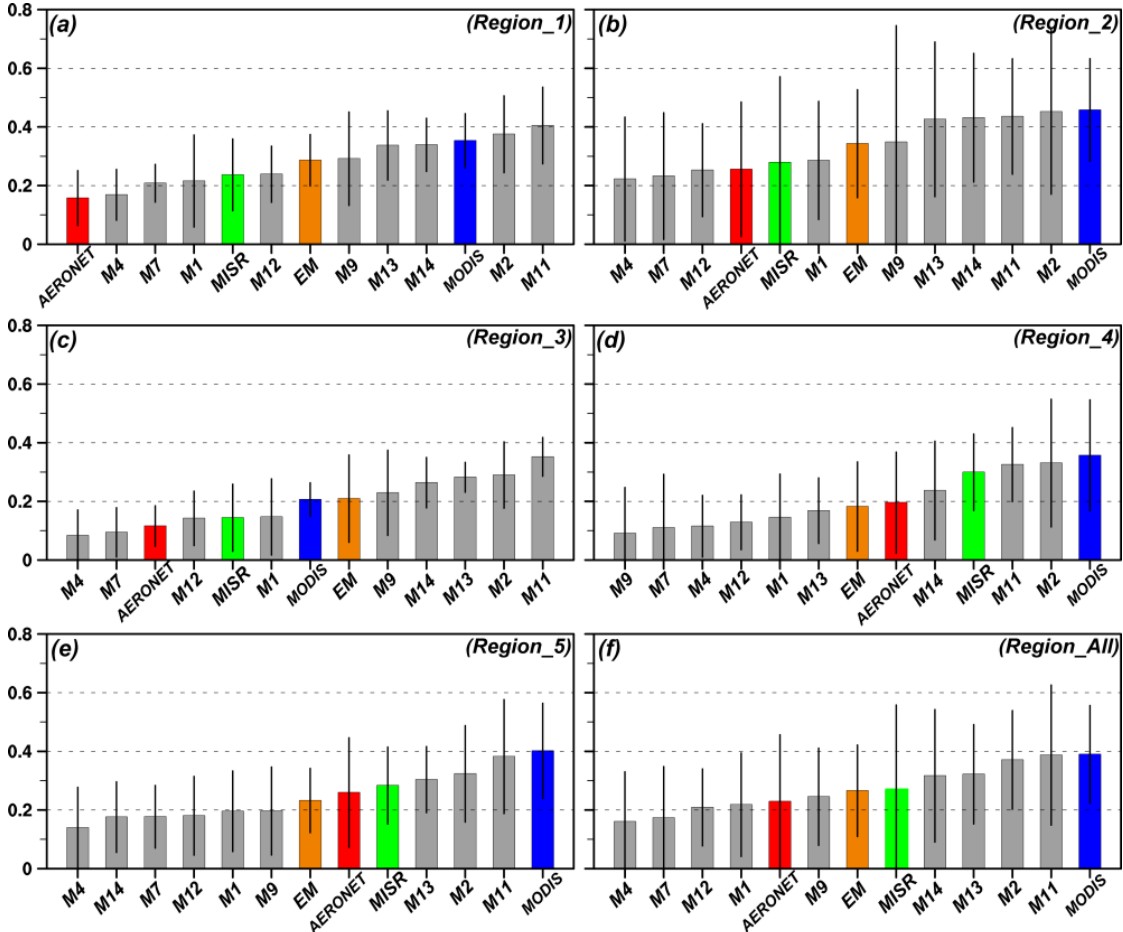

**Figure 11: Multi-model AOD (grey bars) averaged over the five defined regions (Region_1 to Region_5) and the whole analyzed domain (Region_All), together with the ensemble mean predictions (orange bar), measured values retrieved from MODIS (blue bar) and MISR (green bar). AOD averaged over the corresponding AEROENT stations (red bar) in each region are also shown. The error bars represent one standard deviation. (a) Region_1, (b) Region_2, (c) Region_3, (d) Region_4, (e) Region_5, and (f) Regioin_All.**





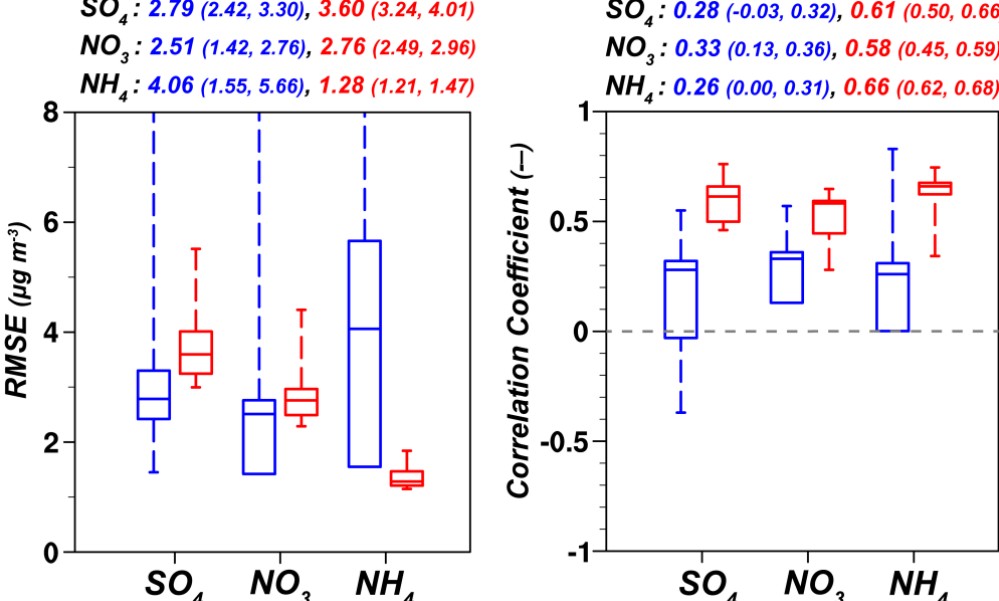

**Figure 12: Intercomparison of model performance in MICS-Asia II (blue) and MICS-Asia III (red) for $SO_4^{2-}$, $NO_3^-$ and $NH_4^+$.**
**Eight models participated in MICS-Asia Phase II. Detailed information can be found in Hayami et al. (2008). Twelve models are**
**analyzed in MICS-Asia Phase III. Statistics (e.g. RMSE and R) are calculated from all the available models against monthly**
**observations provided by EANET. Each boxplot summarizes the statistical information including the interquartile range, the full**
**range and the median. Detailed values of medians (interquartile ranges) for $SO_4^{2-}$, $NO_3^-$ and $NH_4^+$ are also listed at the top of**
**each panel.**





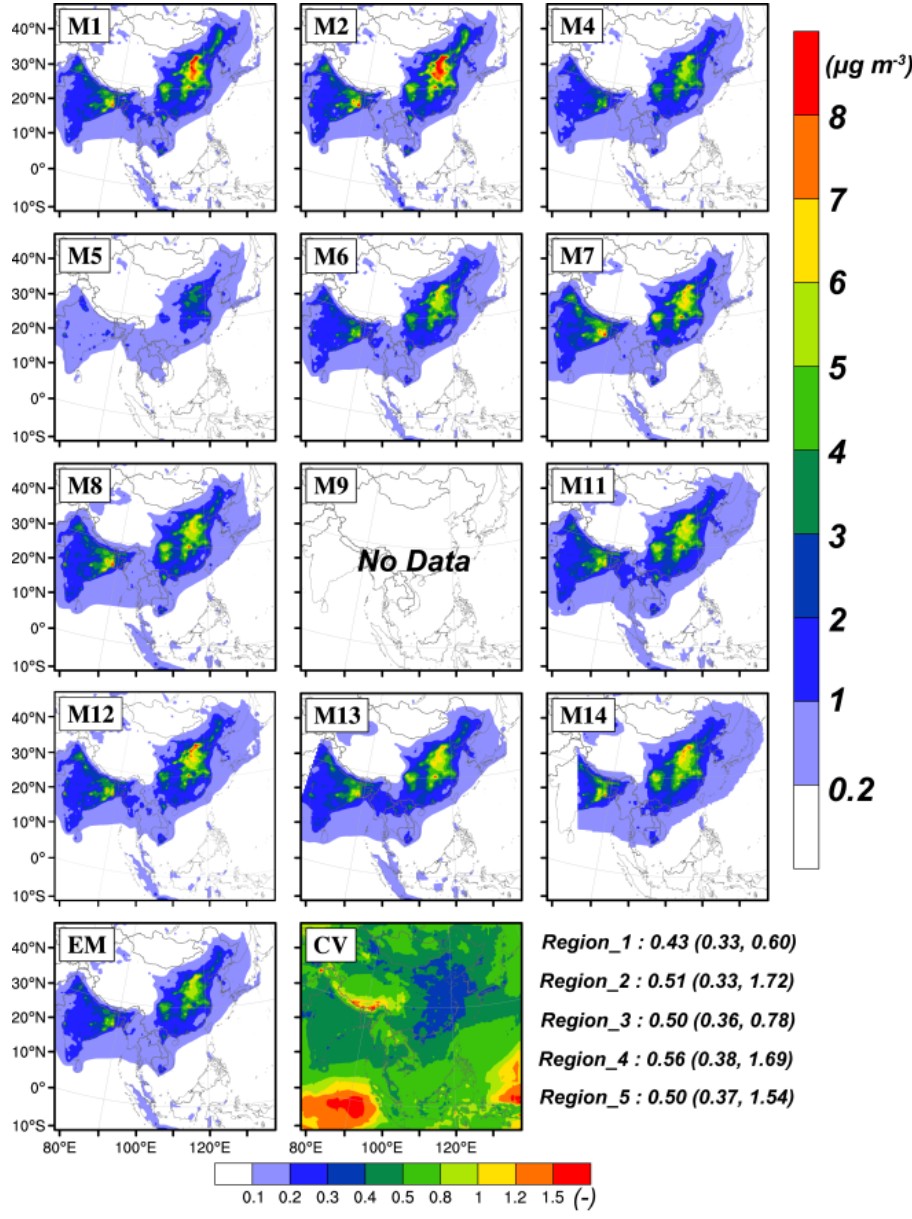

**Figure 13: Spatial distribution of simulated BC concentrations from each participating model and the multi-model ensemble mean. The coefficient of variation (CV), defining as the standard deviation of these models divided by their mean, is also calculated. The values listed in the bottom right corner represent the averaged CV (the minimum CV, the maximum CV) in each defined region.**



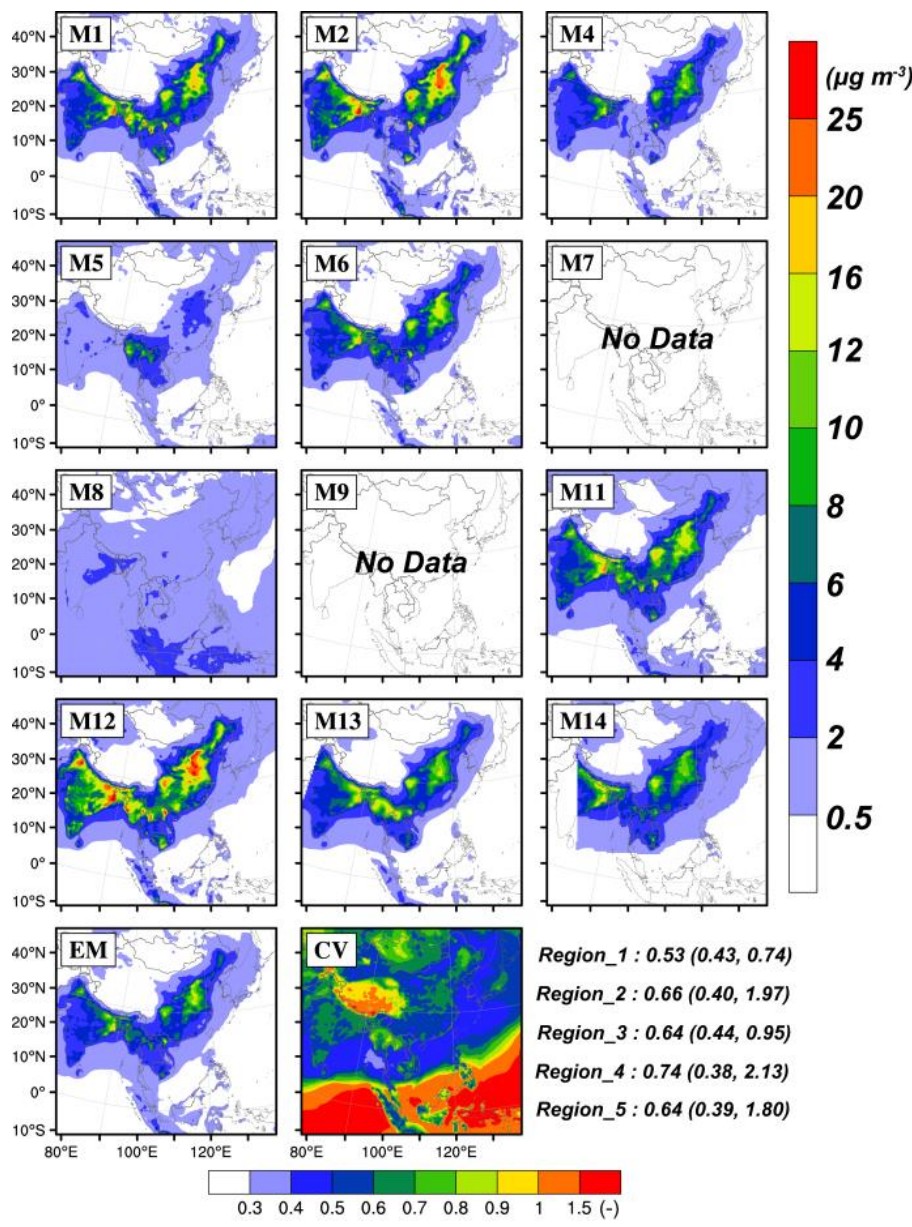

**Figure 14: Similar as Figure 13, but for OC.**





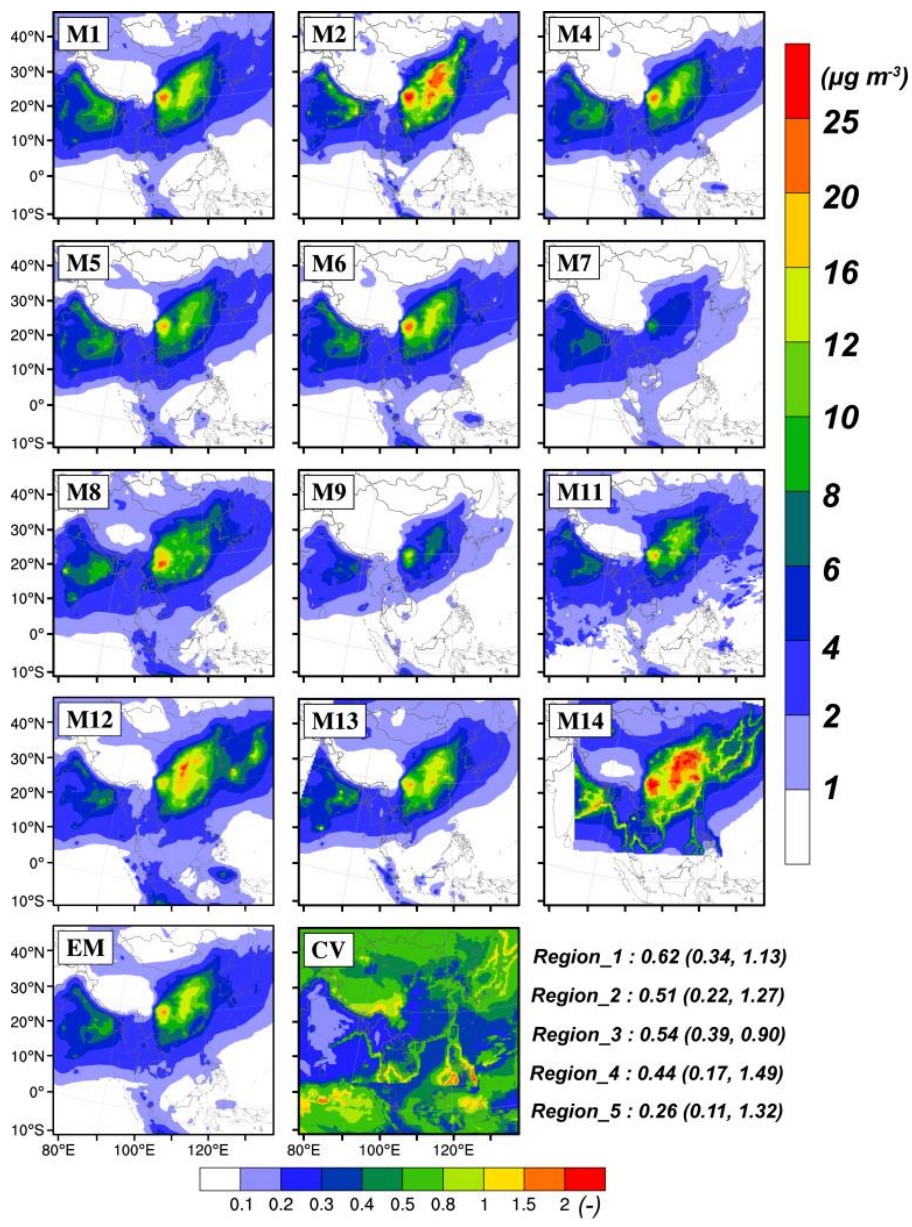

**Figure 15: Similar as Figure 13, but for $SO_4^{2-}$.**





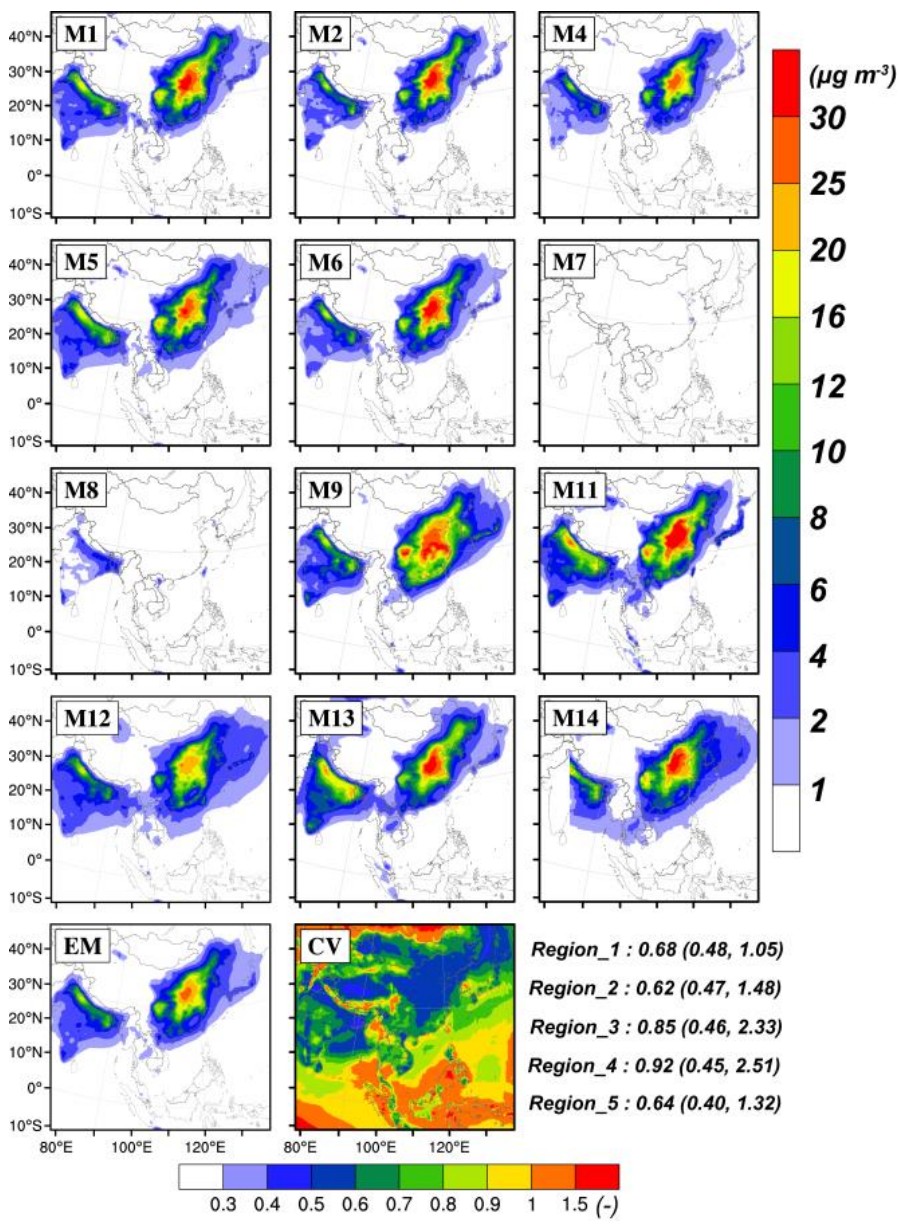

**Figure 16: Similar as Figure 13, but for $NO_3^-$.**





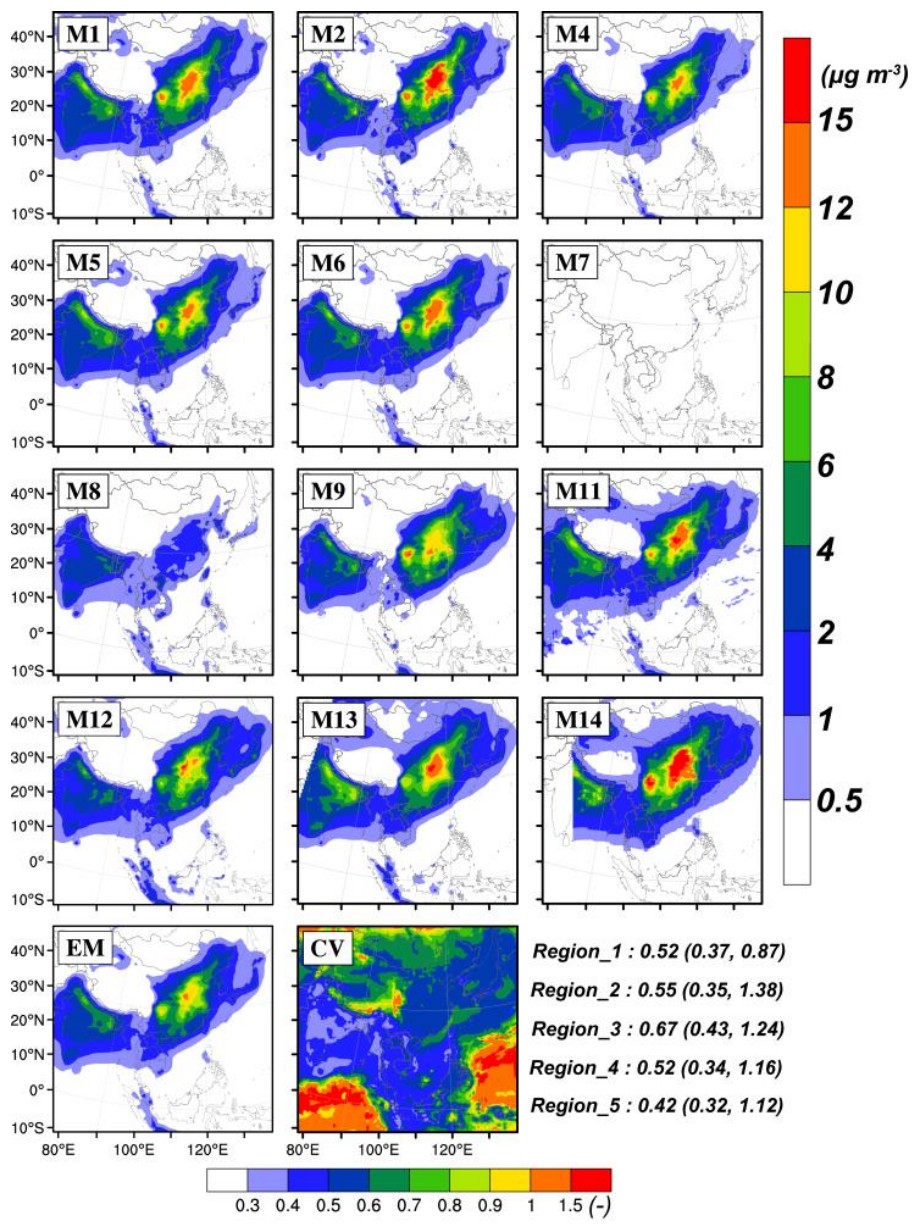

**Figure 17: Similar as Figure 13, but for $NH_4^+$.**





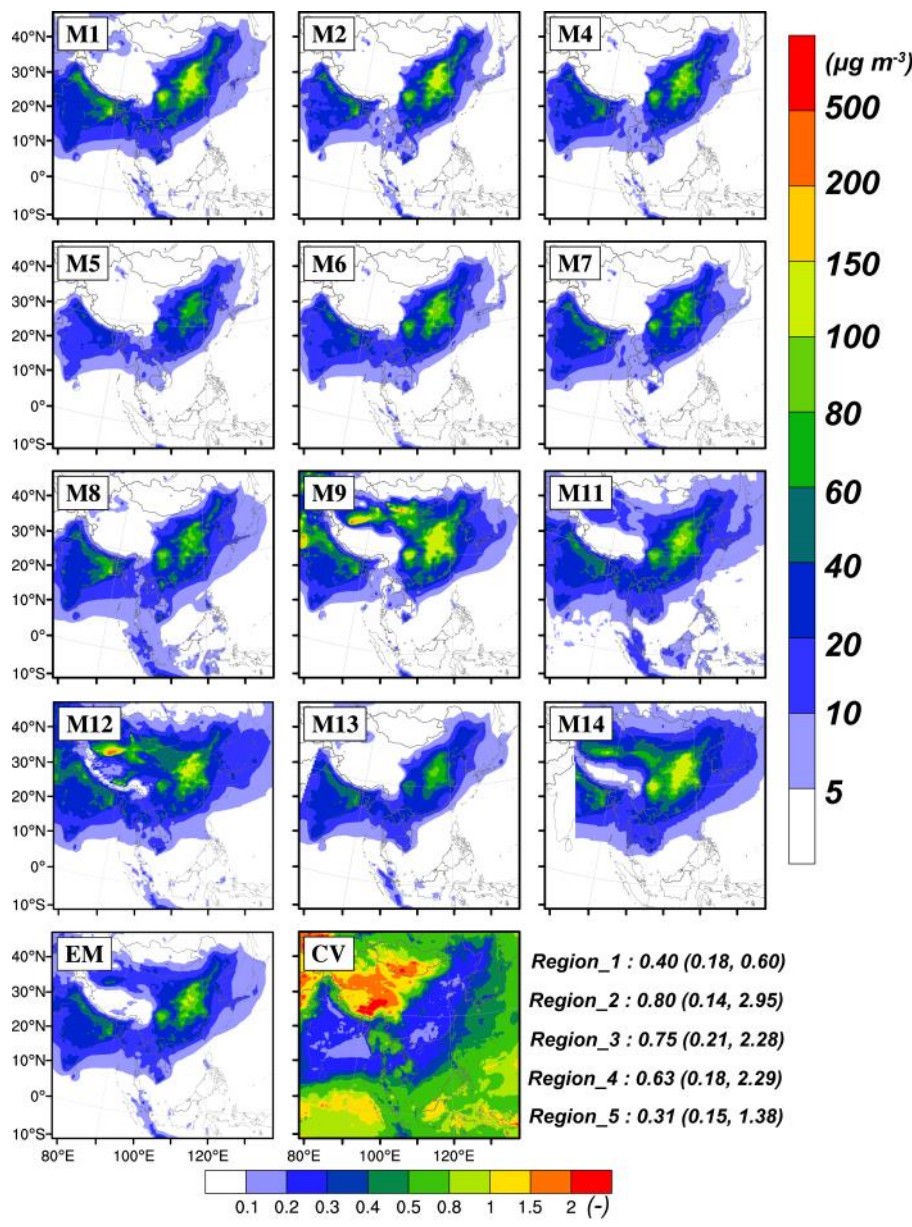

**Figure 18: Similar as Figure 13, but for PM$_{2.5}$.**

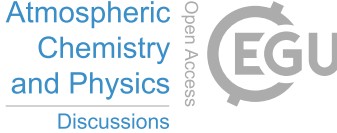



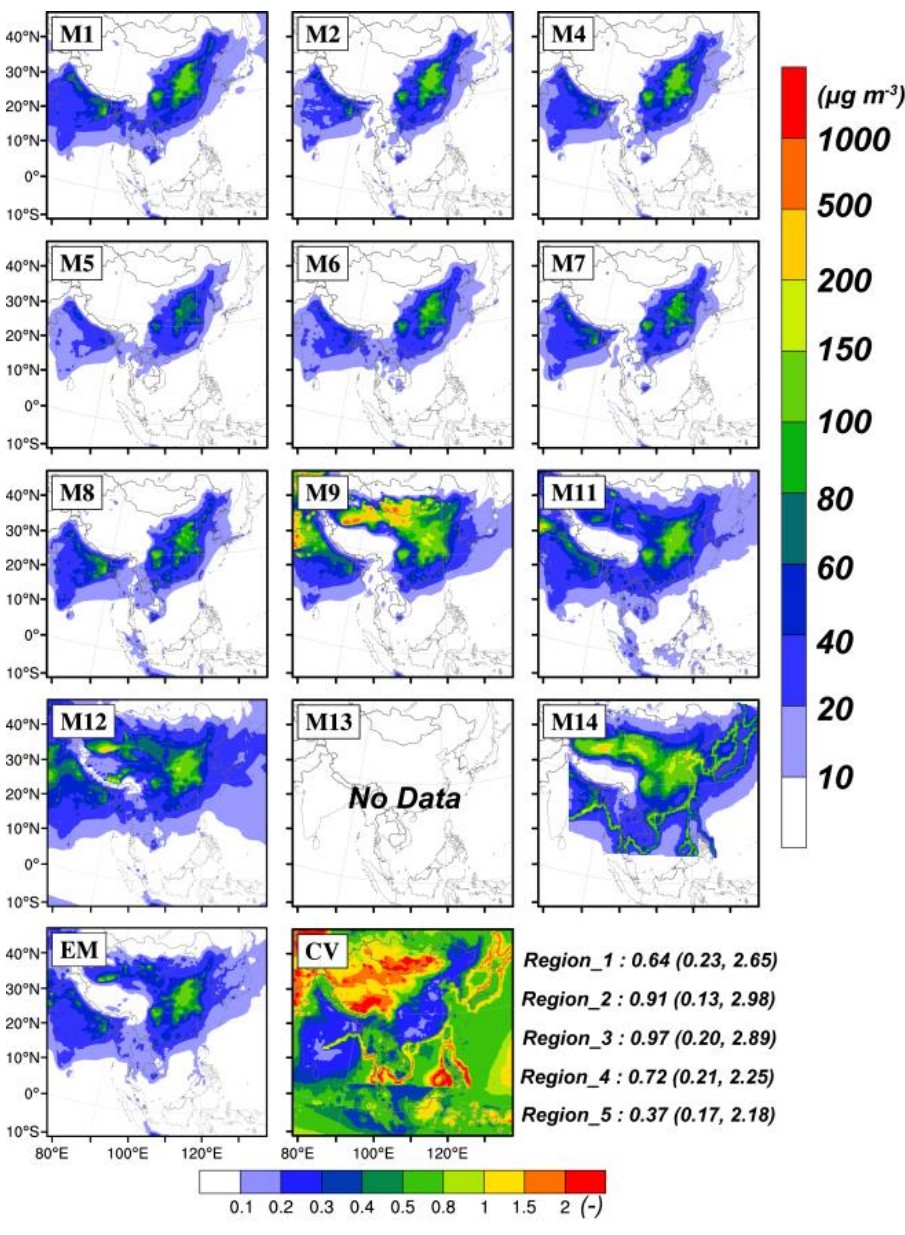

**Figure 19: Similar as Figure 13, but for PM$_{10}$.**



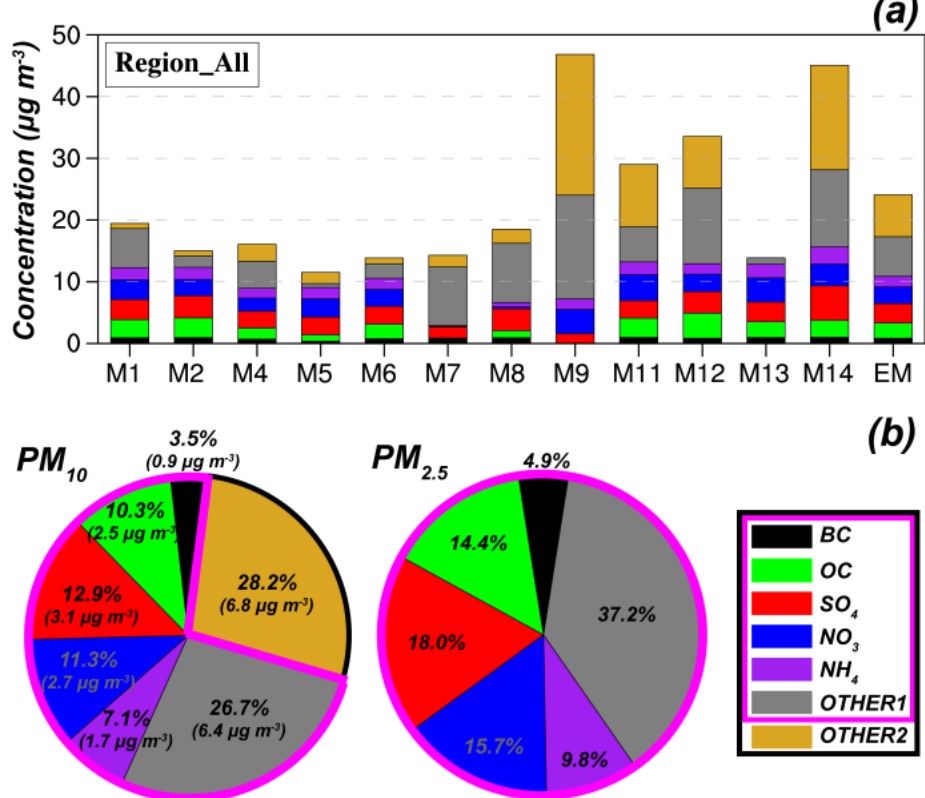

**Figure 20: (a) Chemical compositions of simulated particle matter (PM) from all participating models and the ensemble mean. (b) The ratio of each composition to PM$_{10}$ and PM$_{2.5}$ from multi-model ensemble mean. PM$_{10}$ includes PM$_{2.5}$ and OTHER2, while PM$_{2.5}$ is composed of BC, OC, SO$_4^{2-}$, NO$_3^-$, NH$_4^+$ and OTHER1. Notably, OTHER2 cannot be calculated in M13 because PM$_{10}$ concentration has not been submitted. OC is not available in M7, so we leave it into OTHER1. BC and OC are not available in M9 and these concentrations are grouped into OTHER1.**





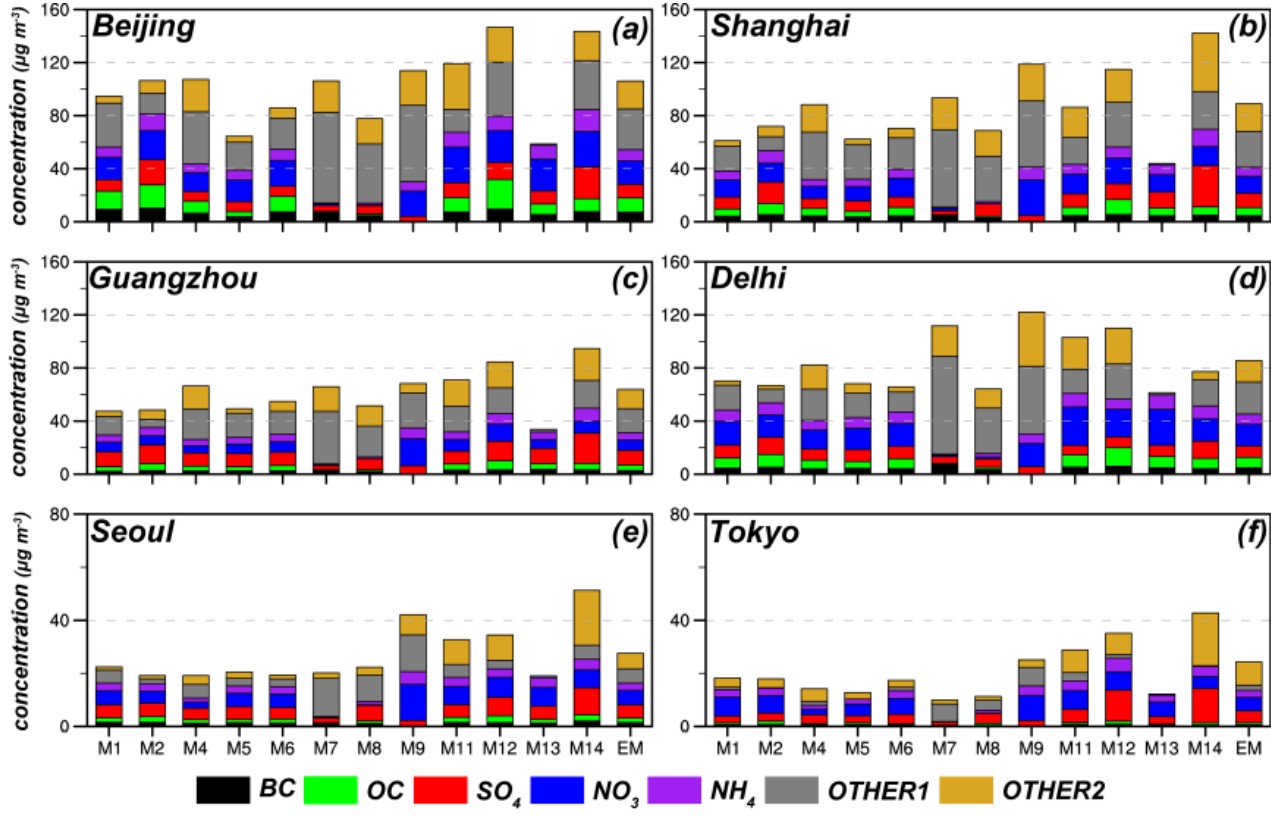

**Figure 21: Chemical compositions of simulated particle matter in six metropolitans. (a) Beijing, (b) Shanghai, (c) Guangzhou, (d)**
**Delhi, (e) Seoul and (d) Tokyo.**





**Figure 22: Ratios of chemical compositions to PM₁₀ (a1-a6) and PM₂.₅ (b1-b6) from multi-model ensemble mean in six cities. Seasonal variations of PM₂.₅ concentrations are also shown (c1-c6).**