# Peer review of "MICS-Asia III: Multi-model comparison and evaluation of aerosol over East Asia"

_Atmospheric Chemistry and Physics, 2018_

## Referee Comment (RC1) · Anonymous Referee #1 · 20 Mar 2019

General comments: East Asia is always undertaken serious haze pollutions in recent years with rapid population and economic growths. And aerosols have significant influences on the air qualities, human health and climate changes through their direct and indirect affections on solar radiation and atmospheric chemistry. The chemical transport models (CTMs) have become critical tools and widely used to address the properties of atmospheric aerosols and their impacts. In this study, 14 CTMs are participate in the MICS–Asia Phase III to evaluate their ability in simulating aerosol species and to document similarities and differences among model performances, also to reveal the characteristics of aerosol chemical components over big cities in East Asia. The topic of this study is interesting and novel to some degrees. And the paper has a potential for publication in the journal after revisions.

Major comments: 1. Aerosols in East Asia are complex in their compositions and temporal-spatial variations. As a more and more important component of the particles, secondary organic aerosol is not taken into account in CTMs, which might lead to the underestimation of PM2.5, PM10 or AOD. 2. Are the natural aerosols such as sea salt and dust included in the simulations? Similar to SOA, dust aerosol also plays an important role in regional air quality in East Asia. If the natural aerosols have been taken into account, what kinds of emission mechanisms are used? 3. A full name is needed when the abbreviation appears in the first time, such as some chemical species and statistical words in Abstract. 4. Is the resolution 45 km accurate enough for air quality simulation? Why not using the nesting framework in some important regions in East Asia. 5. What kinds of methods are used when investigating the ensemble means of the multi-model values? Just averaged from the 14 models or others? 6. In Results and Conclusions, it would be better if the authors could also quantify or highlight the differences among the results from the same mode but with different inputs or physical/chemical processes. 7. Similar to my last comments, in Results and Conclusions, the authors should also quantify or highlight the differences among the results from the different models with the same or different inputs/physical/chemical processes. 8. All kinds of observed sites can be plotted in one panel with different markers and colors in Figure 2 to make it more readable. 9. There are too many figures in the manuscript. The authors can delete some similar figures. 10. Conclusions should be shortened and more concise. 11. English should be improved substantially throughout the whole manuscript.

---

## Referee Comment (RC2) · Anonymous Referee #2 · 14 Apr 2019

Chemical transport models play important roles in advancing understanding of aerosol pollution and aerosol climate effects. This manuscript evaluates multiple model applications in Asia using observations from multiple platforms. The manuscript needs major revisions before publication.

There are two major issuesïijŽ 1 Improvements in language are needed. I would suggest that the authors ask native speakers for help.

2 The authors fail to gain insights out of the evaluation and model inter-comparisons. As the result, the abstract and summary parts are a little weak. More efforts are needed to understand the details of model inputs, reactions, and etc.

Specific comments: 1 Page 4 line 23: to present and summary the: summary should

be summarize; please improve the language carefully through the entire manuscript.

2 Page 7 line 3: weird expression "incredible" here

3 Page 11 line 7: "incorrect treatments of the NH3 emission inputs": this statement is not supported by any evidence in the manuscript. How about plotting NH3 emissions from these two models? From Fig. 15, predicted sulfate from M7 and M8 look consistent with others. If NH3 emissions are not treated well, it should affect sulfate significantly. My sense is that nitrate from M7 and M8 are problematic. Please figure out the real reason.

4 Many statements in the manuscript were presented without showing any evidence. Another example is in page 10 line 27: I doubt M7 and M9 include heterogeneous uptake of SO2 on aerosols. Please make sure the descriptions of model cover the inclusion of important chemical reactions, which will help understand the reasons for differences.

5 What can we learn from the evaluation and comparison? The authors need to add more discussions on this.
* * *

---

## Author Comment (AC1) · 4 Jun 2019

The comment was uploaded in the form of a supplement:
https://www.atmos-chem-phys-discuss.net/acp-2018-1346/acp-2018-1346-AC1-supplement.pdf
* * *

---

## Author Comment (AC2) · 4 Jun 2019

The comment was uploaded in the form of a supplement:
https://www.atmos-chem-phys-discuss.net/acp-2018-1346/acp-2018-1346-AC2-supplement.pdf

---

## Author Response (AR2)

**Response to Comments of Reviewer #1**

**(comments in *italics*)**

**Manuscript number:** acp-2018-1346

**Title:** MICS-Asia III: Multi-model comparison and evaluation of aerosol over East Asia

**General comments:**

*East Asia is always undertaken serious haze pollutions in recent years with rapid population and economic growths. And aerosols have significant influences on the air qualities, human health and climate changes through their direct and indirect affections on solar radiation and atmospheric chemistry. The chemical transport models (CTMs) have become critical tools and widely used to address the properties of atmospheric aerosols and their impacts. In this study, 14 CTMs are participate in the MICS–Asia Phase III to evaluate their ability in simulating aerosol species and to document similarities and differences among model performances, also to reveal the characteristics of aerosol chemical components over big cities in East Asia. The topic of this study is interesting and novel to some degrees. And the paper has a potential for publication in the journal after revisions.*

**Response:**

Thanks to the reviewer for the valuable comments and suggestions which are very helpful for us to improve our manuscript. We have revised the manuscript carefully, as described in our point–to–point responses to the comments.

**Major comments:**

1. *Aerosols in East Asia are complex in their compositions and temporal-spatial variations. As a more and more important component of the particles, secondary organic aerosol is not taken into account in CTMs, which might lead to the underestimation of $PM_{2.5}$, $PM_{10}$ or AOD.*

**Response:**

We totally agree with the reviewer. Air quality models have underestimated the concentrations of secondary organic aerosol (SOA) in both urban and rural areas (Huang et al., 2014; Pye et al., 2015; Woody et al., 2016). This is because many important SOA precursors are not considered in emissions (Carlton et al., 2010). As Gao et al. (2018) pointed out that even though the same emission inventories were used in chemical transport models (CTMs), disparities could also be found in predicted concentrations of organic carbon (OC). This inconsistency may be mainly caused by the different treatments of SOA production in CTMs, including different formation pathways and different empirical parameters (Carlton et al., 2010).

Analyzing the aerosol chemistry mechanisms used in the fourteen CTMs in the first topic of the Model Inter−Comparison Study for Asia (MICS−Asia) Phase III, SOA yield parameterizations in CMAQ (M1–M6, M14), NAQPMS (M11) and NHMChem (M12) are treated by AERO5/6, more details can be found in Edney et al. (2007), Carlton et al. (2010) and Appel et al. (2017). In M7 and M9, the organic chemistry is based on SORGAM (Secondary Organic Aerosol Model) (Schell et al.,

2001). In M8, the volatility basis−set (VBS) approach (Donahue et al., 2006) is used to represent the wide range of the volatility of organic compounds and complex processes. In GOCART (Goddard Chemistry Aerosol Radiation and Transport) aerosol scheme in M10, 10% of organic compounds from the volatile organic compound (VOC) emission inventory are assumed to be converted to SOA (Chin et al., 2002). The formation of SOA in the GEOS−Chem model (M13) is predicted based upon rate constants and aerosol yield parameters determined from laboratory chamber studies (Seinfeld and Pankow, 2003). These different SOA chemistry parameterizations can result in large variations in simulated OC concentrations (Fig. R1), and the domain−averaged CV (coefficient of variation, defined as the standard deviation of the models divided by their mean) can be as high as ~0.65.

[Figure]

**Figure R1.** Spatial distributions of simulated organic carbon (OC) concentrations from each participant model and the MMEM. The calculated coefficient of variation (CV, standard deviation divided by the mean) is also shown. The values listed in the bottom right corner of the figure represent the averaged CV (the minimum CV, the maximum CV) in each defined sub-region.

The large differences in predicted OC concentrations will lead to significant biases in PM$_{2.5}$ concentrations. As shown in Fig. 4(b1) in the revised manuscript, nearly all participant models underestimate the observed PM$_{2.5}$ concentrations in Region_1, with normalized mean bias (NMB) of −39.0% for multi−model ensemble mean (MMEM). This negative bias can also be found in Ikeda et al. (2013), who compared simulation results from CMAQ (v4.7.1) against observations from the same remote stations (Rishiri and Oki) used in this manuscript. Ikeda et al. (2013) pointed that the underestimation of organic aerosols could explain the underpredicted particulate matter concentrations.

[Figure]

**Figure 4.** Time series of the monthly observed and simulated aerosol compositions: (a1) BC, (b1)-(b3) PM$_{2.5}$, (c1)-(c4) PM$_{10}$. The thin grey lines represent simulation results, and the grey shaded areas indicate the spread. The thick black lines are the ensemble mean. The red solid lines mean the observations, and the dashed red lines represent one standard deviation. Correlation coefficients (Rs, shown in black) for the whole year and normalized mean biases (NMBs, shown in blue) for each season between observations and MMEM are shown in each panel. The number of monitoring sites used to calculate the statistics in each sub-region is also listed above each panel. In this figure, the monthly observations except BC are taken from EANET and CNEMC; the monthly BC concentrations are collected from published literatures.

2. *Are the natural aerosols such as sea salt and dust included in the simulations? Similar to SOA, dust aerosol also plays an important role in regional air quality in East Asia. If the natural aerosols have been taken into account, what kinds of emission mechanisms are used?*

**Response to the question "Are the natural aerosols such as sea salt and dust included in the simulations":**

Both the impacts of dust aerosols and sea salts are considered in M9–M14. Sea–salt emissions

are considered in M1–M6, but the windblown dust parameterizations are turned off. Neither the impacts of dust nor sea salt are considered in M7 and M8. More detailed model configurations are listed in new Table 1 in the revised manuscript.

**Response to the question "what kinds of emission mechanisms are used":**

For dust emissions, dust aerosols in M10 and M13 are simulated by the GOCART model (Ginoux et al., 2001); a simplified dust emission parameterization proposed by Shao (2001) is used in M9 (Shao, 2004); a size–segregated dust deflation module proposed by Wang et al. (2000) is used in M11; an empirical dust emission mechanism based on the approach of Gillette and Passi (1988) is applied in M12 and M14 (Han et al., 2004). However, dust schemes in all the WRF–CMAQ models (M1–M6) and the two WRF–Chem models (M7 and M8) are turned off.

For sea–salt aerosols, the method of Clarke et al. (2006) is used in M12 to simulate the sea–salt emissions. In other participant models (sea–salt emission is not considered in M7 and M8), sea–salt emissions are simulated online by using the algorithm proposed by Gong et al. (2003).

Following the reviewer's suggestion, we have added new sections (Section 2.1.3 and Section 2.1.4) in the revised manuscript to briefly describe the dust emission mechanisms and sea–salt emission mechanisms.

"Natural emissions of windblown dust have been explicitly parameterized since CMAQ v5 (Foroutan et al., 2017), but all the participated WRF–CMAQ models did not turn this option on, which means dust aerosols were not considered in M1–M6. Meanwhile, the dust scheme in M7 and M8 was also turned off.

Dust particles in M10 and M13 were simulated by the GOCART model (Ginoux et al., 2001). This model includes eight size groups of mineral dust ranging from 0.1 to 10 μm. The emission flux for a size group can be expressed as follows: $F = C \times S \times s_p \times u_{10}^2 \times (u_{10} - u_t), if\ u_{10} > u_t$, where $C$ is a constant with the value of 1 μg s$^2$ m$^{-5}$. $S$ means the probability source function, representing the fraction of alluvium available for wind erosion. $s_p$ is the fraction of each size group within the soil. $u_{10}$ and $u_t$ are the wind speed at 10 m and threshold velocity of wind erosion, respectively.

A simplified dust emission parameterization proposed by Shao (2001) was used in M9 (Shao, 2004). Dust emission in Shao_2004 is proportional to streamwise saltation flux, and the proportionality depends on soil texture and soil plastic pressure. The size–resolved dust flux goes into four size bins, with diameters ranging from 1.95 to 20 μm (Kang et al., 2011). More detail about the dust emission rate and the total dust flux can be found in Shao (2004).

A size–segregated dust deflation module proposed by Wang et al. (2000) was used in M11. It was developed based on three major predictors (friction velocity, surface humidity and dominant weather system), and has been successfully applied in many dust-related simulations (Wang et al., 2002; Yue et al., 2010). The dust flux $F$ is calculated as follows: $F = C \times \frac{\rho_a}{g} \times E \times u^{*3} \times$

$\left(1 + \frac{u_0^*}{u^*}\right) \times \left(1 - \frac{u_0^{*2}}{u^{*2}}\right) \times \left(1 - \frac{RH}{RH_0}\right)$, where $C$ equals to 10$^{-5}$, $\rho_a$ means air density, $g$ is gravitational acceleration. $E$ is the weighting factor, representing the uplifting capability of land surface. $u_0^*$ and $u^*$ are the fraction and threshold friction velocities, respectively. $RH$ and $RH_0$ are relative humidity and threshold relative humidity, respectively. According to soil categories and vegetation coverage, the dust emission intensity was further modified by Luo and Wang (2006).

Four size bins of dust particles ranging from 0.43 to 10 μm were considered in this emission module. Meanwhile, several heterogeneous reactions on dust particles were also considered (Li et al., 2012a).

An empirical dust emission mechanism based on the approach of Gillette and Passi (1988) was used in M12 and M14 (Han et al., 2004). Dust flux can be calculated through the following formula:

$F = C \times u_*^4 \times \left(1 - \frac{u_*}{u}\right) \times (1 - f \times R), if\ u > u_*$, where $u$ and $u_*$ are the friction and the threshold friction velocities, respectively. C is the correction coefficient $(1.4 \times 10^{-15})$. $f$ and $R$ represent the fractional coverage of vegetation and the reduction factor in a model grid. Dust particles with diameters ranging from 0.43 to 42 μm were grouped into 11 bins, with the first eight bins below 11 μm for aerosol sampler, and the additional three bins above 11 μm for larger particles (Han et al., 2004).

Different dust schemes will produce different dust emission fluxes over arid and semi-arid regions (Zhao et al., 2010; Su and Fung, 2015). Several factors, such as potential source regions, threshold friction velocity, size distribution, and other surface and soil–related parameters used in equations can be the primary causes for the inconsistency, and the differences in simulated dust emissions will affect the characteristics of spatial–temporal variations of atmospheric aerosol particles." **(Section 2.1.3 in Page 8, Line 2−32)**

"As one of the major components of primary aerosols, sea–salt aerosols contributes to 20–40% of secondary inorganic aerosols (SIAs) over coastal regions (Liu et al., 2015; Yang et al., 2016). These particles can provide surface areas for condensation and reaction of nitrogen and sulfur, making the simulated concentrations of SIAs more accurate (Kelly et al., 2010; Im, 2013).

In M12, the method of Clarke et al. (2006) was used to simulate the sea–salt emissions as follows: $S_{100} = \frac{C_s \times k \times V_{wind} \times h}{A_{avg} \times L + 0.5 \times w_0}$. The sea–salt source function ($S_{100}$) is defined as the number of sea–salt aerosols generated per unit area of ocean surface completely covered by bubbles (100% coverage) per unit time. $C_s$ is the differences of condensation nuclei concentrations collected at 5 m (impacted by breaking waves) and 20 m (background values). $k$ is the multiplier for tower $C_s$ compared to mean profile. $V_{wind}$ means surf zone wind speed. $h$ is the height of plume layer for beach profile. $A_{avg}$ represent mean bubble fractional coverage area between waves. $L$ is the distance wave travels to shore, and $w_0$ is the initial width of breaking wave bubble front.

In other participating models (sea–salt emission is not considered in M7 and M8), sea–salt emissions were simulated online by using the algorithm proposed by Gong et al. (2003). The density function $\frac{dF}{dr}$ (m$^{-2}$ s$^{-2}$ μm$^{-1}$) is calculated as follows: $\frac{dF}{dr} = 1.373 \times u_{10m}^{3.41} \times r^{-A} \times (1 + 0.057 \times r^{3.45}) \times 10^{1.607e^{-B^2}}$, where $u_{10m}$ is the 10 m wind speed, $r$ is the particle radius at RH=80%. A represents an adjustment parameter, which control the shape of submicron size distribution. B = $\left(0.433 - log_{10}(r)\right)/0.433$, meaning a parameter related to particle radius. In CMAQ model, the sea–salt scheme was updated by Kelly et al. (2010) to enhance the emission of sea–salt from coastal surf zone, and to allow dynamic transfer of HNO$_3$, H$_2$SO$_4$, HCl, and NH$_3$ between coarse particles and gas phase. In GEOS-Chem model, it was updated by Jaegle et al. (2011) to improve the simulation of sea–salt with dry radii smaller than 0.1 μm." **(Section 2.1.4 in Page 9, Line 2−20)**

3. *A full name is needed when the abbreviation appears in the first time, such as some chemical species and statistical words in Abstract.*

**Response:**

Following the reviewer's suggestion, we have explained all of the abbreviations when they first appear in the revised manuscript.

4. *Is the resolution 45 km accurate enough for air quality simulation? Why not using the nesting framework in some important regions in East Asia.*

**Response to the question "Is the resolution 45 km accurate enough for air quality simulation":**

The objective of MICS–Asia phase III Topic 1 is to evaluate the strengths and weaknesses of current multi–scale air quality models in East Asia applications, including analyzing the similarities and differences between simulation results. For each participant model, a unified simulation domain with 180×170 grid points at 45 km horizontal resolution is requested by organizers in order to reduce the impacts from different model configurations (e.g. grid resolution).

As we all know, model resolution can affect the simulation results, and the influence is more significant for air quality than for meteorological fields (Tan et al., 2015). Li et al. (2016) and Gao et al. (2018) pointed out that a finer resolution could produce smaller NMBs compared with the same model using a larger grid size. However, the requested simulation domain covers a large area (15.4 °S–58.3 °N, 48.5 °E–160.2 °E, including China, Korean Peninsula, Japan, nearly all countries in Southeast Asia, and so on), the finer spatial resolution (< 45 km) will require a tremendous amount of computational cost and data space for all the participant models. Maybe the sensitivity experiments about the model resolutions will be discussed in MICS-Asia Phase IV.

**Response to the question "Why not using the nesting framework in some important regions in East Asia":**

Multi–model estimates of air pollutions by using a nested simulation over haze polluted regions, such as North China Plain, can obtain many robust conclusions about the spatial–temporal variations of aerosols, including the impacts of aerosols. This interesting topic and the interactions between air quality and climate have been discussed in MICS–Asia Phase III Topic 3.

In this manuscript, we mainly focus on the first topic of MICS–Asia Phase III, and try to evaluate the strengths and weaknesses of air quality models by using common meteorological fields, emission data, boundary conditions, and some unified model configurations (e.g. grid resolution).

5. *What kinds of methods are used when investigating the ensemble means of the multi-model values? Just averaged from the 14 models or others?*

**Response:**

Fourteen CTMs (M1–M14) have participated in MICS–Asia Phase III Topic 1, but no data can be acquired from M10, and simulation results in M3 are extremely large. Therefore, multi–model ensemble mean (MMEM) are calculated by averaging all available model results (except M3 and M10). Similar method can also be found in Han et al. (2008) and Gao et al. (2018).

6. *In Results and Conclusions, it would be better if the authors could also quantify or highlight the differences among the results from the same mode but with different inputs or physical/chemical processes.*

**Response:**

According to the model configurations listed in new Table 1 in the revised manuscript, the impacts of boundary conditions (BCs) can be analyzed by comparing the simulation results from M1 and M2. The settings in these two WRF–CMAQ models are similar except the BCs. M1 adopts the downscale results from GEOS–Chem, while M2 uses the default values from CMAQ.

Following the reviewer's suggestion, we have added the following discussions about the impacts of BCs in Section 4 in the revised manuscript: "MICS–Asia project gives an opportunity to understand the performance of CTMs in East Asia applications, including the similarities and differences among air quality models. In order to quantify the impacts of different model inputs and model configurations, and to reduce the diversities among simulation results, more detailed sensitivity experiments should be discussed. For example, simulation results from M1 and M2 can be used to assess the impacts of boundary conditions (BCs), since the configurations in these two models are similar except the BCs. M1 adopts the downscale results from GEOS–Chem, while M2 uses the default values from CMAQ. From Fig. S9 we can find that positive biases are simulated $((M1 - M2)/M2 * 100\% > 0)$, especially around the edges of the simulation domain, and the maximum deviation can be over 100%. This is because the boundary conditions from GEOS-Chem consider the impacts of aerosols outside the domain. All these demonstrate that the impacts of BCs should not be neglected when analyzing the spatial distribution characteristic of simulated aerosols around the edge of the domain. But in most inland regions, differences between M1 and M2 are smaller $(< \pm 10\%)$." **(Section 4 in Page 22, Line 25–34)**

**Table 1. Basic configurations of participant models in MICS−Asia Phase III**

| Model Index | Model Version | Vertical Layers (1st height) | Horizontal advection | Vertical diffusion | Gas phase chemistry | Aerosol chemistry | Dry deposition | Wet scavenging | Dust scheme | Sea-salt scheme | Meteorology | Boundary Condition | Online/Offline | References |
|---|---|---|---|---|---|---|---|---|---|---|---|---|---|---|
| M1 | WRFCMAQ5.0.2 | 40 (57 m) | Yamo | ACM2 | SAPRC99 | Aero6 ISORROPIA(v2) | Wesely | Henry's law | NA | Gong, Kelly | Standard[a] | GEOS-Chem | Online access | Fu et al., (2008a) |
| M2 | WRFCMAQ5.0.2 | 40 (57 m) | Yamo | ACM2 | SAPRC99 | Aero6 ISORROPIA(v2) | Wesely | Henry's law | NA | Gong, Kelly | Standard[a] | Default | Online access | Wang et al., (2014b) |
| M3 | WRFCMAQ5.0.1 | 40 (57 m) | Yamo | ACM2 | CB05 | Aero6 ISORROPIA(v2) | Wesely | Henry's law | NA | Gong, Kelly | Standard[a] | GEOS-Chem | Online access | Lam et al., (2011) |
| M4 | WRFCMAQ4.7.1 | 40 (57 m) | Yamo | ACM2 | SAPRC99 | Aero5 ISORROPIA(v1.7) | Wesely | Henry's law | NA | Gong, Kelly | Standard[a] | CHASER | Offline | Itahashi et al., (2014) |
| M5 | WRFCMAQ4.7.1 | 40 (57 m) | Yamo | ACM2 | SAPRC99 | Aero5 ISORROPIA(v1.7) | M3DRY | Henry's law | NA | Gong, Kelly | Standard[a] | CHASER | Offline | Yamaji et al., (2008) |
| M6 | WRFCMAQ4.7.1 | 40 (57 m) | Yamo | ACM2 | SAPRC99 | Aero5 ISORROPIA(v1.7) | M3DRY | Henry's law | NA | Gong, Kelly | Standard[a] | CHASER | Offline | Nagashima et al., (2017) |
| M7 | WRFChem3.7.1 | 40 (29 m) | 5th order Monotonic | – | RACM−ESR with KPP | MADE/SORGAM | Wesely | Henry's law | NA | NA | WRF/NCEP | Default | Online integrated | Park et al., (2018) |
| M8 | WRFChem3.6.1 | 40 (57 m) | 5th order Monotonic | MYJ | RACM with KPP | MADE/VBS | Wesely | Henry's law | NA | NA | WRF/NCEP | CHASER | Online integrated | Lin et al., (2014) |
| M9 | WRFChem3.6 | 40 (16 m) | 5th order Monotonic | YSU | RADM2 | MADE/SORGAM | Wesely | Henry's law | Shao (2004) | Gong | WRF/NCEP | CHASER | Online integrated | Chen et al., (2017) |
| M10 | NU-WRF v7lis7-3.5.1-p3 | 60 (44 m) | 5th order Monotonic | YSU | RADM2 | GOCART | Wesely | Grell | GOCART | Gong | WRF/MERRA2 | MOZART+GOCART | Online integrated | Tao et al., (2013) |
| M11 | NAQPMS | 20 (50 m) | Walcek and Aleksic (1998) | K−theory | CBMZ | Aero5 ISORROPIA(v1.7) | Wesely | Henry's law | Wang (2000) | Gong | Standard[a] | CHASER | Online access | Wang et al., (2008) |
| M12 | NHMChem | 40 (54 m) | Walcek and Aleksic (1998) | FTCS | SAPRC99 | ISORROPIA(v2) | Kajino | Kajino | Han (2004) | Clarke | JMA NHM | CHASER | Offline | Kajino et al., (2012) |
| M13 | GEOS-Chem9.1.3 | 47 (60 m) | ppm | Lin and McElroy (2010) | Nox-Ox-HC-Br | ISORROPIA(v2) | Wesely | Liu | GOCART | Gong, Jaegle | Geos-5 | NA | Offline | Zhu et al., (2017) |
| M14 | RAMSCMAQ4.6 | 15 (100 m) | Yamo | ACM2 | SAPRC99 | Aero5 ISORROPIA(v1.7) | Wesely | Henry's law | Han (2004) | Gong | RAMS/NCEP | CHASER | Offline | Zhang et al., (2002) |

[a]'Standard' represents the reference meteorological field provided by MICS−Asia III project.

7. *Similar to my last comments, in Results and Conclusions, the authors should also quantify or highlight the differences among the results from the different models with the same or different inputs/physical/chemical processes.*

**Response:**

According to the reviewer's suggestion, a major revision has been made in Section 3.3: Inter–comparison between participant models. Generally, simulation results from all participant models are compared with each other, and the diversities are carefully discussed.

The major results from the inter–model comparison can be summarized as follows:

1. Analyzing the ratio of SNA (sulfate, nitrate and ammonium) to $PM_{2.5}$, large variations are simulated by participant models, with values ranging from 31.1% (M7) to 75.1% (M5). Different gas phase and aerosol schemes used in CTMs can explain this inconsistency.

2. Higher SOR (sulfur oxidation ratio) is calculated by CMAQ models, which means CMAQ may have a more intense secondary formation of $SO_4^{2-}$ than other participant models.

3. Similar NOR (nitric oxidation ration) is predicted by CTMs, but the value (~0.20) is larger than the observed one (~0.15), which means overmuch $NO_3^-$ is simulated by current CTMs.

4. According to the mole ratio of ammonium to sulfate and nitrate, $NH_3$–limited atmospheric condition can be simulated by all participant models, which means a small reduction in ammonia may improve the air quality significantly.

5. The coefficient of variation (CV) can be used to quantify the inter–model deviation, and a large CV is simulated in coarse particles (subtract $PM_{2.5}$ from $PM_{10}$). The poor consistency, especially over the arid and semi–arid regions, is mainly caused by the dust aerosols, which means current CTMs have difficulty in reproducing similar dust emissions by using different dust schemes. But the simulated fine particles are in good agreement among CTMs, especially over the haze–polluted areas.

Detailed descriptions about the comparisons of simulation results from different models with different parameterizations are listed in the revised section **(Section 3.3 in Page 19–21)**. Meanwhile, related conclusions from the inter–model comparisons are also added in Section 4 in the revised manuscript **(Section 4 in Page 21–23)**.

8. *All kinds of observed sites can be plotted in one panel with different markers and colors in Figure 2 to make it more readable.*

**Response:**

Thanks. We have revised the figure (Fig. 2) according to the reviewer's suggestion.

[Figure]

**Figure 2.** The geographical locations of observation stations: EANET (shown in black circles, the number of stations is 39), CNEMC (shown in red triangles, the number of stations is 32), Others (observations collected from published literatures, shown in purple stars, the number of stations is 32), and AERONET (shown in black boxes, the number of stations is 33). Five defined sub-regions (Region_1 to Region_5) are also shown.

9.  *There are too many figures in the manuscript. The authors can delete some similar figures.*
**Response:**

Thank you for your suggestion. In the revised manuscript, only 11 figures are used. Several similar figures are deleted, and some interrelated figures are merged together.

10. *Conclusions should be shortened and more concise.*
**Response:**

Thank you for your suggestion. In the revised manuscript, words about the model descriptions and model evaluations in the discussion part (Section 4) have been cut back by about 40%. But several reasons and evidences have been added in the discussion part to explain the deviations between observations and simulations. These explanations may be helpful for future studies. Conclusions from the revised section of "Inter–comparison between participant models" (Section 3.3) are also briefly summarized, aiming to quantify the differences between simulation results from different models with different parameterizations, including highlighting the common results presented by current CTMs. For example, comparing with other models, higher SOR is shown in CMAQ, which means more intense secondary formation of $SO_4^{2-}$ can be simulated by CMAQ models. Similar NOR is predicted by all participant models, but the value is higher than the observed one, indicating the overestimation of $NO_3^-$ may be a common phenomenon in current CTMs. According to the large CV over arid and semi–arid regions, it still remains challenging to estimate dust emissions by using different dust schemes in current CTMS.

In the discussion part in Section 4, differences among the simulation results from the same model but with different inputs are also analyzed, which may be helpful to reduce the diversities of simulated aerosol concentrations in air quality models.

*11. English should be improved substantially throughout the whole manuscript.*

**Response:**

Thank you for your suggestion. The language in the entire revised manuscript has been carefully corrected.

*two models? From Fig. 15, predicted sulfate from M7 and M8 look consistent with others. If NH3 emissions are not treated well, it should affect sulfate significantly. My sense is that nitrate from M7 and M8 are problematic. Please figure out the real reason.*

**Response:**

Thank you for your suggestion.

Generally, there are two pathways about the $NO_3^-$ formation in air quality models. The dominant pathway is the homogeneous gas−phase reactions between $HNO_3$ and $NH_3$ under ammonia−rich conditions, and the second pathway is the heterogeneous hydrolysis of $N_2O_5$ on aerosol surface at night in ammonia−poor environment. As $H_2SO_4$ is nonvolatile, and the equilibrium surface concentration in the model is set to be zero. So $(NH4)_2SO_4$ is the preferential species in the completion when $H_2SO_4$ and $HNO_3$ are both present, and $NH_4NO_3$ is formed only if excess $NH_3$ is available beyond the sulfate requirement.

However, the mole ratio of $nNH_4^+/nSO_4^{2-}$ (n refers to the molar concentration) calculated by M7 and M8 are relative small (0.42 for M7 and 1.1 for M8), which means acidic sulfate cannot be fully neutralized by ammonia to form $(NH_4)_2SO_4$ or even $NH_4HSO_4$, especially in M7. This is because extremely low concentrations of $NH_3$ are simulated by M7 and M8 (Fig. S4). So fewer $NH_4NO_3$ and/or $NO_3^-$ can be formed. Meanwhile, the hydrolysis of $N_2O_5$ is not considered in M7 and M8. All these result in the lower concentrations of $NO_3^-$.

[Figure]

**Figure S4.** Spatial distributions of simulated NH₃ concentrations from each participant model.

Simulated $NH_4^+$ concentrations are influenced by the partitioning between gaseous $NH_3$ and aerosol $NH_4^+$, and are also associated with the $SO_4^{2-}$ and $NO_3^-$ concentrations. Even though the same $NH_3$ emissions are employed by all participant models (Fig. S2(c)), extremely low values are simulated by M7 and M8, especially over the mainland of China, which means fewer gaseous $NH_3$ can be converted to aerosol $NH_4^+$. Analyzing the mole ratios of $nNH_4^+/nSO_4^{2-}$ from M7 (0.42) and M8 (1.1), lower concentrations of $NH_4^+$ will be simulated by M7.

[Figure]

**Figure S2.** The merged emissions of (a) SO₂, (b) NOₓ, (c) NH₃ and (d) PM₂.₅ in 2010 from MIX (anthropogenic emission), MEGAN (biogenic emission), GFED (biomass burning emission), air and ship emission, and volcanic emission. The unit for gas is Mmol/month/grid, and the unit for particulate is Mg/month/grid.

Following the reviewer's suggestion, we have revised the sentences as "For $NO_3^-$, low concentrations are observed in Region_1 (1.5 μg m⁻³), Region_3 (0.6 μg m⁻³) and Region_4 (1.8 μg m⁻³), but high values are presented in Region_2 (13.4 μg m⁻³), showing the similar spatial distribution characteristics as the observed $SO_4^{2-}$. In CTMs, there are two pathways about the nitrate formation. The dominant pathway is the homogeneous gas−phase reaction between $HNO_3$ ($NO_2$ oxidation by OH during the daytime) and $NH_3$ under ammonia−rich conditions, and the second pathway is the heterogeneous hydrolysis of $N_2O_5$ on aerosol surface at night in ammonia−poor environment (Seinfeld and Pandis, 2006; Archer-Nicholls et al., 2014). As $NH_4NO_3$ is semi−volatile species, and the equilibrium surface concentration of $H_2SO_4$ is set to be zero in CTMs, so $(NH_4)_2SO_4$ is the preferential species in the completion when $H_2SO4$ and $HNO_3$ are both present. Only if $NH_3$

is excess, then $NH_4NO_3$ will been formed. Analyzing the performance of each participant model, $NO_3^-$ concentration is overpredicted by most models, and the underestimation of $SO_4^{2-}$ can be used to explain this overestimation (Chen et al., 2017). Meanwhile, the biases of model calculated gas−phase oxidation (e.g. $NO_2 + OH \rightarrow HNO_3$ ) and/or gas−aerosol phase partitioning (e.g. $HNO_{3(g)} + NH_{3(g)} \leftrightarrow NH_4NO_{3(s, \ aq)}$) may also result in the overestimation (Brunner et al., 2014; Gao et al., 2014). However, M7 and M8 significantly underestimate the observed $NO_3^-$ concentrations (NMB~−93.4%). One reason for the extremely low values may result from the incorrect concentrations of $NH_3$ simulated by M7 and M8 (Fig. S4). As Chen et al. (2016) pointed out that the amount of $NH_3$ is a key factor in determining the $NO_3^-$ concentration. Another reason for this underestimation is M7 and M8 did not consider the impacts of $N_2O_5$ heterogeneous reaction ($N_2O_{5(g)} + H_2O_{(aq)} \rightarrow 2HNO_{3(aq)}$). Su et al. (2017) pointed out that the hydrolysis of $N_2O_5$ can led up to 21.0% enhancement of $NO_3^-$, especially over polluted regions. Although the NMB calculated in Region_All for MMEM is only −1.1%, MMEM systematically overpredicts observations in Region_1 (NMB=45.2%) and Region_3 (NMB=38.2%), but underpredicts in Region_2 (NMB=−0.7%) and Region_4 (NMB=−44.9%)" and "Simulated $NH_4^+$ concentrations are influenced by the partitioning between gaseous $NH_3$ and aerosol $NH_4^+$, and are also associated with the $SO_4^{2-}$ and $NO_3^-$ concentrations (Gao et al., 2018). Model predictions (except M7, M8 and M14) can reproduce the measurements relatively well in each defined sub−region. But significant overestimation is shown by M14, while significant underestimation is simulated by M7 and M8, especially in Region_2 with NMBs of 72.2% for M14, −94.9% for M7, and −81.0% for M8, respectively. For M14, overestimated $SO_4^{2-}$ and $NO_3^-$ make the concentrations of $NH_4^+$ higher, since more ammonium is required to neutralize particle−phase acid. For M7 and M8, extremely low concentrations of $NH_3$ are simulated, which means fewer gaseous $NH_3$ can be converted to aerosol $NH_4^+$. In general, the calculated NMB in Region_All for MMEM is 4.0%." **(Section 3.1.1 in Page 13−14)**

4. *Many statements in the manuscript were presented without showing any evidence. Another example is in page 10 line 27: I doubt M7 and M9 include heterogeneous uptake of SO2 on aerosols. Please make sure the descriptions of model cover the inclusion of important chemical reactions, which will help understand the reasons for differences.*

**Response:**

Thank you for your suggestion. We have revised the sentence: "As Zheng et al. (2015) and Shao et al. (2019) pointed out that missing sulfate formation mechanisms (e.g. heterogeneous sulfate chemistry) on aerosol in current air quality models may result in this underestimation, especially in China where significant increase of secondary aerosols (such as sulfate) can be observed during polluted periods (Liu et al. 2015)." **(Page 13, Line 16−19)**

According to the reviewer's comments, detailed descriptions about the model parameterizations, including gas phase chemistry **(Section 2.1.2.1 in Page 5−6)**, aerosol chemistry **(Section 2.1.2.2 in Page 6−7)**, dust scheme **(Section 2.1.3 in Page 8)** and sea−salt scheme **(Section 2.1.4 in Page 9)**, have been added in the revised manuscript.

5.  *What can we learn from the evaluation and comparison? The authors need to add more discussions on this.*

**Response:**

Thank you for your suggestion. MICS−Asia project gives an opportunity to understand the performance of CTMs in East Asia applications, including the similarities and differences between simulation results from participant models.

From the revised model evaluation section **(Section 3.1, Page 12−18)**, we can conclude that:

(1) Air quality models can well reproduce the spatial and temporal variability patterns of aerosols in East Asia in 2010;

(2) Multi–model ensemble mean (MMEM) shows better performance than most single−model predictions, which means analyzing the simulation results from MMEM can have a relative common understanding of the properties of atmospheric aerosols and their impacts;

(3) The higher uncertainties in emission inventory, the larger biases will be simulated by air quality models;

(4) Nearly all participant models underpredict the concentrations of $SO_4^{2-}$, which means sulfate formation mechanisms should be updated in current CTMs.

From the revised model inter–comparison **(Section 3.3, Page 19−21)**, we can conclude that:

(1) High $PM_{2.5}$ concentrations ($> 40$ μg m$^{-3}$) can be reproduced by all participant models, and the calculated CV (coefficient of variation) averaged over these regions are low ($< 0.3$), indicating similar performance of the air quality models in simulating the air pollutants over haze–polluted areas;

(2) Even though similar magnitude of $PM_{2.5}$ are simulated, the ratio of SNA (sulfate, nitrate and ammonium) to $PM_{2.5}$ varies a lot (about a factor of 2) among participant models, which means different gas–phase and aerosol chemistry mechanisms used in these models cause this inconsistency;

(3) CMAQ models show higher SOR (sulfur oxidation ratio) than other participant models, so more intense secondary formation of $SO_4^{2-}$ is simulated by CMAQ;

(4) Similar NOR (nitric oxidation ratio) is predicted by participant models, but the value (~0.20) is larger than the observed one (~0.15), meaning more $NO_3^-$ is simulated by secondary formation in current air quality models;

(5) $NH_3$–deficient atmospheric condition can be successfully simulated by all participant models. A small reduction in ammonia will make the neutralizing effect weaker, and fewer SNA can be formed, which may improve the air quality significantly.

(6) For coarse particles (subtract $PM_{2.5}$ from $PM_{10}$), large CV is calculated, which means low consistency can be found in the simulation results. The low consistency of simulated coarse particles in each region is mainly caused by the dust aerosols, indicating current air quality models have difficulty in producing similar dust emissions by using different dust schemes.

Meanwhile, following the reviewer's suggestion, we also add the following discussion about the impacts of BCs in the revised manuscript: "MICS–Asia project gives an opportunity to understand the performance of CTMs in East Asia applications, including the similarities and differences among air quality models. In order to quantify the impacts of different model inputs and model configurations, and to reduce the diversities among simulation results, more detailed

[revised manuscript text omitted]

*Correspondence to*: M.G. Zhang (mgzhang@mail.iap.ac.cn)

**Abstract.** Fourteen chemical transport models (CTMs) participate in the topic 1 of the the Model Inter-Comparison Study

for Asia (MICS–Asia) Phase III. These model results are compared with each other and  an extensive set of measurements, aiming to evaluate the current CTMs' ability in simulating aerosol  concentrations,  to document the similarities and differences among model performances,  to reveal the characteristics of aerosol  components in large cities in East Asia. In general, CTMs can well reproduce the spatial distribution and  temporal variability of aerosol s in East Asia during the year 2010, and multi–model ensemble mean (MMEM) shows better performance than most  participant models, with correlation coefficients ranging from 0.65 ($NO_3^-$) to 0.83 ($PM_{2.5}$). But the concentrations of black carbon, $SO_4^{2-}$ and $PM_{10}$ are underestimated by MMEM, with normalized mean biases (NMBs) of −17.0%, −19.1% and −32.6%, respectively. Positive biases are simulated in $NO_3^-$ (NMB=4.9%), $NH_4^+$ (NMB=14.0%) and $PM_{2.5}$ (NMB=4.4%). In comparison with the statistics calculated from MICS–Asia Phase II, frequent updates of chemical mechanisms in CTMs during recent years make the inter–model variability of simulated aerosol concentrations smaller, and better performance can be found in reproducing the variation tendency of observations. However, a large variation (about a factor of 2) of the ratios of SNA (sulfate, nitrate and ammonium) to $PM_{2.5}$ is calculated among participant models, and a relative more intense secondary formation of $SO_4^{2-}$ is simulated by CMAQ models due to the higher SOR (sulfur oxidation ration) than other models (0.51 vs. 0.39). Similar NOR (nitric oxidation ratio) is predicted by CTMs, but the value is large (~0.20), indicating overmuch $NO_3^-$ is produced by current models. $NH_3$–limited condition can be reproduced by all participant models (the mole ratio of ammonium to sulfate and nitrate is smaller than 1), and a small reduction in ammonia may improve the current air quality. ~~Underestimations of BC (NMB=−17.0%), $SO_4^{2-}$ (NMB=−19.1%) and $PM_{10}$ (NMB=−32.6%) are simulated by EM, but positive biases are shown in $NO_3^-$ (NMB=4.9%), $NH_4^+$ (NMB=14.0%) and $PM_{2.5}$ (NMB=4.4%). Simulation results of BC, OC, $SO_4^{2-}$, $NO_3^-$ and $NH_4^+$ among CTMs are in good agreements, especially over polluted areas, such as the eastern China and the northern part of India. But large coefficients of variations (CV > 1.5) are also calculated over arid and semi–arid regions. This poor consistency among CTMs may attribute to their different processing capacities for dust aerosols.~~ A large coefficient of variation (CV>1.0) is shown in simulated coarse particles, especially over arid and semi–arid regions, which means current CTMs have difficulty in estimating similar dust emissions by using different dust schemes. According to the simulation results in the six Asian cities from MMEM, different air–pollution control plans should be made due to their different major air pollutants in different seasons. MICS–Asia project gives an opportunity to understand the performance of air quality models in East Asia. In order to acquire a mature comprehension of the properties of atmospheric aerosols and their impacts, and to reduce the diversities of simulated aerosols among CTMs, more detailed sensitivity experiments about parameterizations and model inputs should be carried out in future.

[revised manuscript text omitted]
 tThe seasonalityy of observed and simulated aerosol particle mass concentrationsaerosol compositions, including BC, $SO_4^{2-}$, $NO_3^-$, $NH_4^+$, PM2.5 and PM10, are shown in Fig. 4 and Fig. 5. According to the defined sub−regions as illustrated in Fig. 2, all simulations and observations are grouped into the five regions, All simulations and observations are grouped into five defined regions as illustrated in Fig. 3, with the modeling results sampled at the corresponding observation sites stations before averaging together. Individual models are represented shown by the thin grey lines, with the grey shaded areas indicating their spread. The thick black line is represent the MMEM, Tthe red solid line is the observational mean, and the dashed red lines represent represent one standard deviation for each group of stations. In each panel, Tthe correlation coefficients (Rs) for MMEMs versus the monthly observations are calculated in each panel, and and the normalized mean biases (NMBs) in each season (spring: from March to May; summer: from June to August; autumn: from September to November; winter: January, February and December) for EM are also given.

15  The measured BC concentrations in Region_2 exhibit an obvious seasonal variation, with the minimum (~ 3.5 μg m⁻³) during in spring and summer, and the maximum (~ 8 μg m⁻³) during late autumn and winter. All participating Participant models can capture this observed seasonality quite well, and nearly all modeling simulation results are within the standard deviation of the observations, but a large inter−model variation is foundalso simulated, especially in winter when BC concentration is high. Due to its low reactivity in the atmosphere, this variation may be caused by their simulated
20  meteorological conditions, including the impacts of different coupling ways between meteorological and chemical modules (Gao et al., 2015b). Different coupling ways between meteorological and chemical modules, as listed in Table 1, can be used to explain this variation. As Gao et al. (2015b), Briant et al. (2017) and Huang et al. (2018) concluded that the online integrated models can simulate higher BC concentrations than offline models, especially during polluted periods. The correlation coefficient for EM MMEM is 0.73.

25  In each monthFor PM2.5, the mean observedobserved monthly PM2.5 concentrations over in Region_2 is are larger higher than that those in Region_1. This is because the emissions of primary aerosols and their precursors in China are larger than that in Japan and Korean Peninsula (as shown in Fig. S12). But Nnearly all models tend to underpredict the magnitude concentrations of PM2.5 
[revised manuscript text omitted]

High values of PM$_{2.5}$ and PM$_{10}$ in Beijing, Shanghai, Guangzhou and Delhi can be simulated by nearly all models, and the annual mean concentrations of PM$_{2.5}$ and PM$_{10}$ from MMEM are all larger than the IT-1 (Interim target-1, 35 μg m$^{-3}$ for PM$_{2.5}$, 70 μg m$^{-3}$ for PM$_{10}$) proposed by WHO. But relative small concentrations are presented in Tokyo (15.5 and 21.3 μg m$^{-3}$ for PM$_{2.5}$ and PM$_{10}$, respectively) and Seoul (21.7 and 27.6 μg m$^{-3}$ for PM$_{2.5}$ and PM$_{10}$, respectively). For each city, a large spread of concentrations of aerosol compositions can be found among participant models (a factor of ~10 for SNA, a

factor of ~2 for $PM_{2.5}$ and $PM_{10}$). This is partly caused by the differences in gas–aerosol partitioning and dust emissios, including the removal processes (e.g. dry and wet depositions).

~~From Fig. 20 we can find that the simulated concentrations of $PM_{10}$ vary a lot by about a factor of 4 among models, with the highest in M9 (46.5 μg m⁻³) and the lowest in M5 (11.5 μg m⁻³). This large spread can be explained by the differences in simulated concentrations of OTHER2, which is mainly composed of dust aerosol and sea salt aerosol. Generally, the mean $PM_{10}$ concentration from EM is 24.1 μg m⁻³, including 0.9 μg m⁻³ (3.5%) for BC, 2.5 μg m⁻³ (10.3%) for OC, 3.1 μg m⁻³ (12.9%) for $SO_4^{2-}$, 2.7 μg m⁻³ (11.3%) for $NO_3^-$, 1.7 μg m⁻³ (7.1%) for $NH_4^+$, 6.4 μg m⁻³ (26.7%) for OTHER1 and 6.8 μg m⁻³ (28.2%) for OTHER2. For $PM_{2.5}$, the regional mean concentration from EM is 17.3 μg m⁻³, with an inter–model range from 9.7 μg m⁻³ of M5 to 28.1 μg m⁻³ of M14. Except OTHER1, the major compositions in $PM_{2.5}$ in East Asia are $SO_4^{2-}$ (18.0%), $NO_3^-$ (15.7%) and OC (14.4%).~~

~~Aerosol chemical compositions in six high–profile cities in East Asia (Beijing, Shanghai, Guangzhou, Delhi, Seoul and Tokyo) simulated by each participating model and the multi–model EM are shown in Fig. 21. High values of $PM_{2.5}$ and $PM_{10}$ in Beijing, Shanghai, Guangzhou and Delhi can be simulated by nearly all models, while relative small concentrations are presented in Seoul and Tokyo. For each city, a large spread of PM concentrations can be found among models, this is mainly caused by the differences of the simulated concentrations of OTHER1 and OTHER2. In other words, although common emissions are used, different physical–chemical parameterizations can cause large uncertainties in transmission and remove processes of aerosols, including the emission processes of dust and sea salt.sPM ($PM_{10}$ andMM22and Tokyo. Ain $PM_{2.5}$ (Fig. 22(b1–b6)), except OTHER1componentone~~ species in Guangzhou (22.2%). Similar contributions of $SO_4^{2-}$ and $NO_3^-$ can be found in Shanghai, Seoul and Tokyo. All these suggest that different air–pollution control plans should be made in different metropolitans.

For seasonal variations of $PM_{2.5}$ concentrations (Fig. 11(C1–C6)), the highest values in Beijing (107.6 μg m⁻³), Shanghai (87.5 μg m⁻³), Guangzhou (59.9 μg m⁻³) and Delhi (108.7 μg m⁻³) are all simulated in winter. This can be explained by their high emissions during this season. However, in Tokyo, the highest $PM_{2.5}$ concentration  is in summer (21.8 μg m⁻³) and the lowest value is in winter (10.3 μg m⁻³). In Seoul, $PM_{2.5}$ concentrations are comparable during the four seasons.

**4 Conclusion and Discussion**

This manuscript mainly focuses on the first topic of the MICS–Asia Phase III, and intends to
* * *
*Margin annotations (tracked-change callouts):*

analyze the following  objectives: (1) provide a comprehensive evaluation of  current  air quality models against observations, (2) analyze the diversity of simulated aerosol$s$  among participant models, and (3) reveal the characteristics of  aerosol components in the high–profile cities in East Asia.

~~Fourteen regional modeling groups participating in Topic 1 are required to simulate aerosol species using common meteorological fields, emission inventories and boundary conditions during the entire year of 2010 in East Asia. Model predictions are compared with each other, and with measurements of BC, OC, $SO_4^{2-}$, $NO_3^-$ $NH_4^+$, $PM_{2.5}$ and $PM_{10}$. Aerosol optical depth is also rigorously evaluated against observations from AERONET, MODIS and MISR. Note that all simulation results from M3 are incredible, and no data is gained from M10. Meanwhile, M5, M6 and M8 did not submit simulated AOD. M13 did not submit simulated $PM_{10}$. M7 did not submit OC. Neither BC nor OC was submitted from M9.~~

Comparison$s$$s$ against monthly observations from EANET and CNEMC demonstrate that all participant models can well reproduce the spatial and temporal variability patterns in aerosols,  and multi–model ensemble mean (MM) shows better performance than most models, with Rs ranging from 0.65 ($NO_3^-$) to 0.83 ($PM_{2.5}$) . Significant biases  between  predictions and observations can also be found , such as $SO_4^{2-}$ is underestimated by participant models (except M12 and M14) with NMBs ranging from −67.7% to −1.6%, while most models overestimate the concentrations of $NO_3^-$ and $NH_4^+$, and the NMBs are 4.9% and 14.0% for MM, respectively. The absence of sulfate formation mechanisms (e.g. heterogeneous chemistry) in CTMs can explain the underestimation of $SO_4^{2-}$, and the underestimated $SO_4^{2-}$ will result in the overestimation of $NO_3^-$. However, significant underestimations of $NO_3^-$ and $NH_4^+$ are shown in M7 and M8. This is because extremely low values of $NH_3$ are simulated by these models.  The inter–model spread of simulated $PM_{2.5}$ is large, with NMBs ranging from −26.5% of M13 to 46.0% of M14, and nearly all models underestimate the $PM_{2.5}$ concentrations in Region_1. This is because the precursors and the formation pathways of organic aerosols are insufficient in current CTMs, which may cause this negative bias.  
[revised manuscript text omitted]
** *(5.0 µg m⁻³)* *(nstd=5)* | R | 0.70 | 0.73 | 0.71 | 0.65 | 0.70 | 0.73 | **0.80** | – | 0.69 | 0.68 | 0.75 | 0.72 | 0.73 |
| | NMB(%) | **1.0** | 12.7 | −24.7 | −54.9 | −17.8 | −11.7 | −34.2 | – | −17.5 | −2.2 | −26.8 | −11.6 | −17.0 |
| | RMSE | 4.10 | 4.30 | 2.95 | 4.06 | 2.99 | 2.69 | 2.84 | – | 2.91 | 3.52 | 2.80 | **2.64** | 2.77 |
| **$SO_4^{2-}$** *(3.8 µg m⁻³)* *(nstd=31)* | R | 0.69 | 0.71 | 0.64 | 0.58 | 0.66 | 0.48 | 0.53 | 0.65 | 0.55 | 0.50 | **0.76** | 0.46 | 0.69 |
| | NMB(%) | −23.1 | −13.0 | −31.0 | −26.4 | −26.9 | −67.7 | **−1.6** | −67.0 | −34.5 | 23.2 | −31.9 | 69.3 | −19.1 |
| | RMSE | 3.21 | **3.00** | 3.46 | 3.57 | 3.35 | 4.64 | 3.62 | 4.45 | 3.78 | 4.01 | 3.24 | 5.51 | 3.22 |
| **$NO_3^-$** *(1.7 µg m⁻³)* *(nstd=31)* | R | 0.55 | 0.51 | 0.62 | **0.65** | 0.58 | 0.45 | 0.29 | 0.64 | 0.59 | 0.60 | 0.43 | 0.58 | **0.65** |
| | NMB(%) | 9.0 | −7.2 | −42.7 | **−1.7** | −11.8 | −81.2 | −80.6 | 125.7 | 46.5 | 54.0 | 22.7 | 35.4 | 4.9 |
| | RMSE | 2.70 | 2.71 | 2.48 | 2.29 | 2.46 | 3.37 | 3.18 | 4.37 | 2.89 | 2.80 | 2.96 | 2.62 | **2.27** |
| **$NH_4^+$** *(1.1 µg m⁻³)* *(nstd=31)* | R | 0.67 | 0.64 | 0.68 | 0.66 | 0.69 | 0.55 | 0.34 | **0.75** | 0.66 | 0.62 | 0.64 | 0.68 | 0.71 |
| | NMB(%) | 23.2 | 33.7 | −10.6 | **7.4** | 14.6 | −93.5 | −34.2 | 45.3 | 35.0 | 49.9 | 34.9 | 56.3 | 14.0 |
| | RMSE | 1.24 | 1.42 | 1.15 | 1.21 | 1.16 | 1.83 | 1.53 | 1.26 | 1.27 | 1.54 | 1.29 | 1.47 | **1.11** |
| **PM$_{2.5}$** *(51.4 µg m⁻³)* *(nstd=14)* | R | 0.80 | 0.78 | 0.80 | 0.71 | 0.80 | 0.80 | 0.77 | 0.82 | 0.80 | 0.78 | 0.75 | 0.81 | **0.83** |
| | NMB(%) | 10.0 | 13.6 | **−1.3** | −25.3 | −5.8 | −5.7 | −15.3 | 26.2 | 5.2 | 31.4 | −26.5 | 46.0 | 4.4 |
| | RMSE | 27.56 | 34.88 | 23.03 | 28.00 | 21.80 | 23.54 | 24.83 | 28.52 | 22.06 | 34.87 | 27.10 | 35.85 | **21.23** |

[revised manuscript text omitted]

**Figure 9. Spatial distributions of simulated PM2.5 concentrations from each participant model and the MMEM. The calculated coefficient of variation (CV, standard deviation divided by the mean) is also shown. The values listed in the bottom right corner of the figure represent the averaged CV (the minimum CV, the maximum CV) in each defined sub-region. The ratio of SNA (sulfate, nitrate, and ammonium) to PM2.5, the SOR (sulfur oxidation ratio), the NOR (nitric oxidation ratio), and the PNR (particle neutralization ratio) are also given at the bottom of each panel.**

[Figure]

[Figure]

**Figure 10. The same as Figure 9, but for PMcoarse (coarse particles, subtract PM2.5 from PM10).**

[revised manuscript text omitted]